# Evaluating Newtonian Mechanics in Video Generative Models with Real Physical Systems

**Antonios Tragoudaras** [* 1]  **Chenyu Zhang** [* 1 2]  **Daniil Cherniavskii** [* 1]  **Antonios Vozikis** [† 1]  **Thijmen Nijdam** [† 1]  **Derck W. E. Prinzhorn** [1]  **Mark Bodracska** [1]  **Nicu Sebe** [2]  **Andrii Zadaianchuk** [‡ 1]  **Stratis Gavves** [‡ 1]

## Abstract

Recent advances in image and video generation raise hopes that these models possess world modeling capabilities—the ability to generate realistic, physically plausible videos. This could revolutionize applications in robotics, autonomous driving, and scientific simulation. However, before treating these models as world models, we must ask: Do they adhere to physical laws? Current evaluation methods rely on subjective judgments or trajectory matching, limiting their usage for physical reasoning estimation, where many generations could be physically plausible. Thus, we introduce **Morpheus**, one of the first physics-informed evaluation frameworks for measuring the ability of video generation models to comprehend Newtonian dynamics. **Morpheus** features 130 real-world videos capturing physical phenomena, guided by conservation laws. Using those as conditioning for video generation, we assess physical plausibility leveraging interpretable metrics evaluated with respect to infallible conservation laws known per physical setting, leveraging advances in physics-informed neural networks and vision-language foundation models. Importantly, **Morpheus** targets controlled Newtonian rigid-body settings to enable quantitative checks. Our findings reveal that even with advanced prompting and video conditioning, contemporary models struggle to encode physical principles despite generating aesthetically pleasing videos. Code and data available here.

---

[*]Equal contribution (first-authors). [†]Equal contribution (second authors). [‡]Equal advising. [1]University of Amsterdam, Amsterdam, The Netherlands [2]University of Trento, Trento, Italy. Correspondence to: Antonios Tragoudaras .

*Proceedings of the 43^{rd} International Conference on Machine Learning*, Seoul, South Korea. PMLR 306, 2026. Copyright 2026 by the author(s).

## 1. Introduction

Video generative models (VGMs) such as SORA (Brooks et al., 2024), COSMOS (Agarwal et al., 2025), and Veo3 (Veo-Team et al., 2024) have improved rapidly, building on remarkable advances in image generative models (Ramesh et al., 2021; Saharia et al., 2022; Podell et al., 2023; Yu et al., 2023), and achieving unprecedented levels of visual fidelity and realism. These developments have motivated growing interest in video generative models as potential *world models* (Puspitasari et al., 2024; Agarwal et al., 2025). In this context, a world model is capable of understanding and predicting the dynamics, causal interactions, and underlying mechanisms of the physical world.

Studying and evaluating physical dynamics goes beyond both human judgment and language-based validation, which assess plausibility by answering questions such as *"does this video of an object falling look legitimate?"* (Bansal et al., 2025; Meng et al., 2025). While such evaluations are useful for coarse screening, they are inherently subjective and cannot probe whether a model captures the underlying physical mechanisms or respects unobserved physical constraints such as mass, inertia, or force coupling. Evaluating physical dynamics in video also goes beyond purely visual or geometric verification (Agarwal et al., 2025), which checks temporal consistency of appearances or trajectories (Kang et al., 2025; Motamed et al., 2026). Visual coherence alone does not guarantee physical correctness: a generated trajectory may look smooth and plausible, yet violate physical laws due to inconsistent initial conditions or incorrect coupling between hidden parameters and dynamics. In many cases, physically meaningful deviations—arising, for instance, from mass distribution, air resistance, or internal structure—are subtle and visually underdetermined, making them difficult to detect through appearance-based metrics alone.

To rigorously evaluate video generation as world models, we must move beyond pixel-level similarity and instead assess whether generated videos respect physical invariants. This is essential particularly in embodied AI settings where agents must act reliably in the real world (Black et al., 2025; Abeyruwan et al., 2025; Bjorck et al., 2025; Kim et al.,

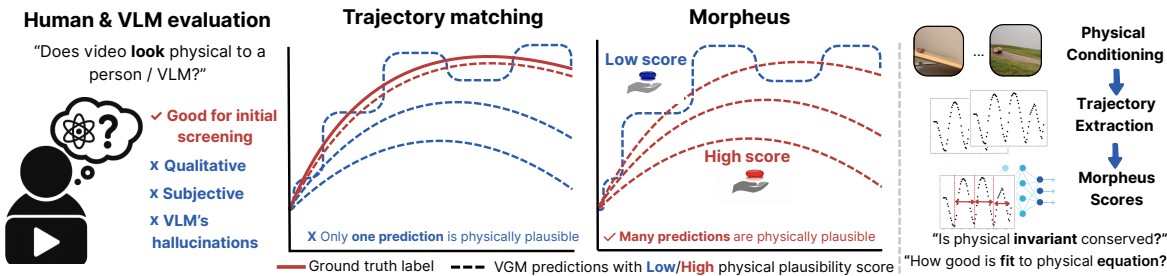

*Figure 1.* Comparison of evaluation methods for video generative models. a) Human or VLM-based judgments provide only qualitative and subjective assessments of physical plausibility. b) Trajectory matching compares generated and ground-truth paths but may misclassify physically valid trajectories. For example, for projectile motion, many parabolic trajectories are physically plausible when VGMs are conditioned only on an image, as object velocity cannot be estimated from it. c) Our proposed framework, Morpheus, evaluates generated videos via physics-informed scores, testing both conservation of physical invariants and consistency with governing equations of motion.

2026; Mao et al., 2025). Physical invariants are quantities derived from system trajectories that govern system dynamics and evolve according to well-defined laws. For example, in Newtonian systems, object trajectories must be consistent with Newton's equations of motion, and conserved quantities such as energy or momentum impose strong constraints on admissible dynamics. Quantifying such invariants enables principled, quantitative benchmarks that probe whether VGMs capture underlying physical dynamics rather than merely producing visually plausible motion.

To this end, we focus on Newtonian mechanics and leverage the power of physics-informed machine learning (Raissi et al., 2017; Karniadakis et al., 2021) – originally proposed for modeling fully observable scientific data – and re-purpose them to fit physical dynamics in noisy video data. We target Newtonian mechanics for two reasons. The primary reason is that Newtonian mechanics govern the majority of physical dynamics in robotic and embodied environments (Abeyruwan et al., 2025; Yang et al., 2024b; Barcellona et al., 2025). A second practical reason is that underlying Newtonian mechanics are deterministic and can be largely estimated from video in controlled settings, thus making it easier to optimize physics-informed neural networks. This is in contrast to chaotic or highly non-linear types of physics that are harder to visually quantify, such as fluid dynamics, electromagnetics, thermodynamics, or optics.

Our contributions are twofold. (i) We introduce a physics-informed evaluation framework that provides a suite of interpretable, physics-based metrics for systematically assessing physical reasoning in video generative models. The framework extracts latent state variables from generated videos and evaluates them against governing ordinary differential equations (ODEs) and conservation laws. (ii) We conduct a comprehensive evaluation of state-of-the-art video generative models on **Morpheus**, including closed-source models such as Veo3 (Veo-Team et al., 2024) and Kling (Kling AI, 2024), as well as open-source models including

Wan2.1 (Wang et al., 2025), COSMOS (Agarwal et al., 2025), CogVideoX (Yang et al., 2025), PyramidalFlow (Jin et al., 2025), and LTX-Video (HaCohen et al., 2024). By guiding model generations using frames from our curated real-world initial conditions, we demonstrate that while leading models achieve impressive visual fidelity, they consistently fall short in capturing real-world physical dynamics.

## 2. Related work

**Evaluation of VGMs** Benchmarking video generation models has evolved to include comprehensive evaluation frameworks that assess multiple dimensions of video quality, temporal coherence, and alignment with prompts. Approaches such as EvalCrafter (Liu et al., 2024b), VBench (Huang et al., 2024), VBench++ (Huang et al., 2025), AIGCBench (Fan et al., 2024), and TC-Bench (Feng et al., 2024) emphasize diverse metrics to evaluate visual fidelity, motion smoothness, spatial consistency, and temporal dynamics. For example, EvalCrafter (Liu et al., 2024b) uses metrics like Motion-Aware Consistency (MAC) and Scene Change Consistency (SCC) to assess the smoothness and natural progression of motion, while VBench introduces metrics for spatial relationships and subject identity consistency to evaluate logical scene composition. Despite the breadth of these, they concentrate on perceptual and semantic aspects of video generation, whereas **Morpheus** focuses on physical plausibility of the generated videos.

**Physical reasoning and plausibility in VGMs** Recent advances in evaluating physical plausibility in video generation have employed both human assessments (Bansal et al., 2025) and automated approaches leveraging Vision-Language Models (VLMs) (Bansal et al., 2025; Meng et al., 2025) as well as object tracking metrics (Wang et al., 2024b; Motamed et al., 2026; Agarwal et al., 2025) (see Table 2 for comparison). Notable frameworks include VideoCon-Physics (Bansal et al., 2025), PhyGenBench (Meng et al., 2025) and PhysBench (Chow et al., 2025), which uti-

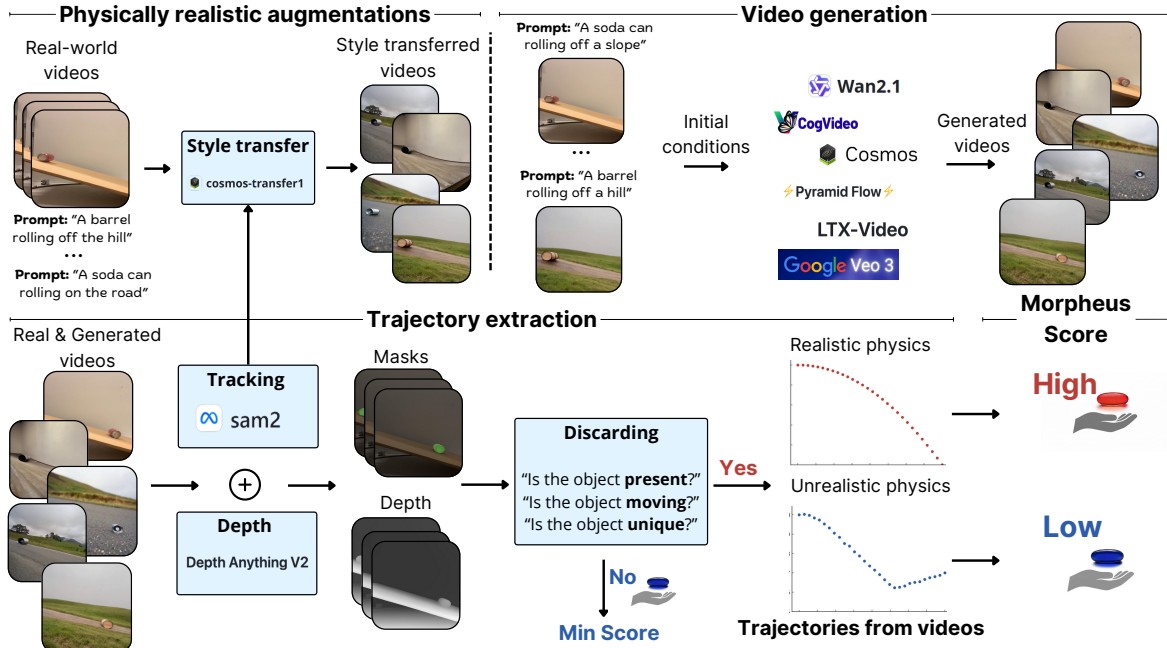

*Figure 2.* The overview of **Morpheus**. Video augmentation and generation (upper) and the trajectory extraction pipelines (lower). We start with augmenting recorded videos with realistic style transfer, based on object masks. Next, we use the first frame (or multiple frames in case of video conditioning) of the obtained videos, as well as the textual description, as a prompt for a VGM. After this, we extract object trajectories for both real-world and generated videos using the trajectory extraction pipeline, including trajectory tracking and discarding unreliable trajectories. Finally, we evaluate **Morpheus** scores for all videos with valid trajectories.

lize VLMs to assess adherence to physical law prompts; VAMP (Wang et al., 2024b), which quantifies motion characteristics through acceleration and velocity variance; and Physics-IQ (Motamed et al., 2026) and COSMOS (Agarwal et al., 2025), which compare object masks between generated and real-world videos. LikePhys (Yuan et al., 2026) introduces a training-free evaluation by leveraging the model's internal denoising score objective to design an ELBO-based likelihood surrogate to distinguish between valid/invalid physical pairs. While this effectively avoids visual appearance biases, it evaluates the model's internal probability distribution rather than the physical fidelity of its generated outputs.

Despite addressing diverse physical phenomena, such approaches suffer significant limitations. VLM and human evaluations often identify physical deviations categorically, like noting gravity violations without quantifying them. Moreover, VLM can hallucinate (Li et al., 2023) and miss subtle physical inconsistencies (Chow et al., 2025). On the other side, object tracking metrics are often based on simulated data (Agarwal et al., 2025; Bakhtin et al., 2019) and assume that modeled processes should be deterministic and predictable (Kang et al., 2025; Motamed et al., 2026).

Crucially, current evaluation paradigms often fail to bridge the gap between video generation and actionable world modeling. Newtonian dynamics represent, arguably

one of the most fundamental layer of world modeling required for embodied agents, table-top manipulation, and Vision-Language-Action (VLA) systems (Black et al., 2025; Abeyruwan et al., 2025; Bjorck et al., 2025). While coverage to domains like fluids or optics is still valuable, prioritization of depth and controllability in this fundamental domain is essential for Robot Learning. **Morpheus** is designed to ensure that physics-informed metrics are interpretable and verifiable in this regime, providing a solid foundation for evaluating models intended to serve as simulators for physical interaction.

**Learn physical invariants and equations from data** Significant progress has been made in learning conservation laws from trajectories (Liu & Tegmark, 2021), and discovering equations in hybrid dynamic systems (Liu et al., 2024c). Mechanistic Neural Networks (MechNN) (Pervez et al., 2024) are able to learn governing ODEs from data, while Mechanistic PDE Networks (Pervez et al., 2025) can learn Partial Differential Equations (PDEs). Conversely, to compare the theoretical prediction with input data, PINNs (Cuomo et al., 2022; He et al., 2023), which integrate physical equations in the loss function, help identify possible physical factors causing errors (such as unmodeled friction, air drag, etc) because it is able to learn corrections to make the predictions closer to the actual observed values.

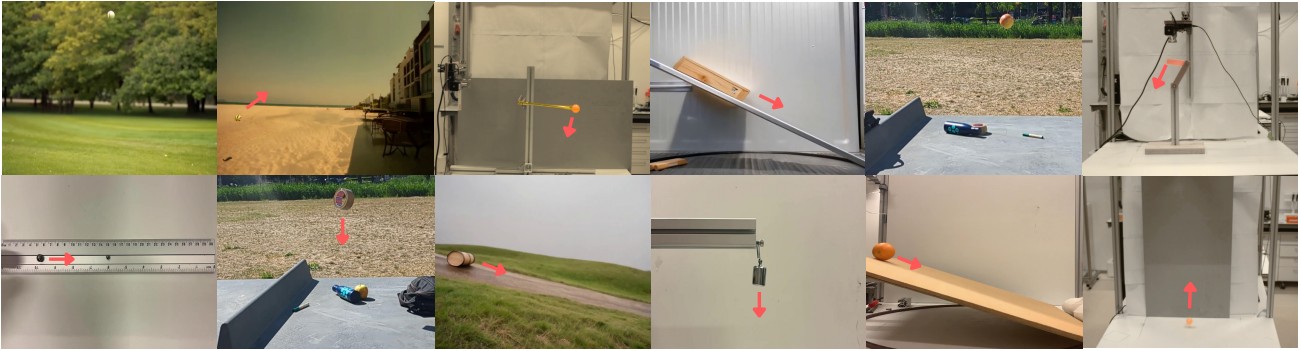

*Figure 3.* Examples of physical experiments included in the **Morpheus** physics-informed evaluation framework, illustrating both different dynamics and variations in object types. Top row (left to right): falling ball, projectile motion, holonomic pendulum, sliding, falling apple, and double pendulum. Bottom row (left to right): collision, falling tape, rolling barrel, spring, rolling orange, and bouncing.

## 3. Morpheus Physics-informed Evaluation

To rigorously examine discrepancies in the physical laws exhibited by generated videos, we propose the **Morpheus** physics-informed evaluation framework. It consists of a dataset for controlled conditioning, evaluation scores, and an analysis of the performance of state-of-the-art models.

**Dataset methodology**   We create a dataset of real-world videos of specific physical phenomena, focusing on capturing fundamental aspects of Newtonian mechanics. Videos are recorded under controlled conditions, allowing us to systematically vary initial parameters and capture repeatable scenarios. By operating in a controlled and repeatable setting, we can isolate and test adherence to specific physical laws – such as the periodic dynamics of a harmonic pendulum – rather than merely assessing overall visual plausibility. This sets our dataset apart from previous works that often focus on uncontrolled, general-purpose video data (Motamed et al., 2026; Kang et al., 2025), allowing for a more precise and targeted evaluation of physical consistency.

We perform a total of about 130 physical experiments, repeating them 10–20 times under different physical initializations. Each physical experiment is governed by different physical principles. For example, gravity influences falling objects, projectile motions, and bouncing; periodic movements influence spring and holonomic pendulum dynamics; friction and normal forces influence inclined sliding, rolling, and more complex dynamical systems such as multi-body collisions and double pendulum (See Fig. 3 and App. A for dataset illustration and description). The varying initial conditions include, for instance, the velocity for falling, the angle and velocity for projectile motions, or the angle for pendulums. Our diverse initializations allow for a robust coverage of dynamic behaviors, and a thorough evaluation of generated videos against real-world physical phenomena. We use the initial video frame(s) to condition the VGM's sampling and generation process. By doing so, we ensure

that the generated sequences start from the same conditions as the real experiments.

**Visually diverse conditioning for robust VGMs evaluation**   To obtain more robust evaluations, it is important to study the performance of VGMs under diverse natural initializations of the same physical process. This reduces sensitivity to the specific setup of the originally recorded experiments. Thus, we augment the initial videos with visually diverse yet reliably generated variants. In particular, we use the video-to-video transfer method (Alhaija et al., 2025) with object masks extracted from recorded videos to obtain diverse and visually realistic videos following an additional text prompt for adaptation. While changing the semantics and appearance of the objects, the generated videos remain constrained by the provided object masks and are reliable augmentations for conditioning in image-to-video and video-to-video generation. To ensure high quality, we filter augmented initializations through manual inspection for visual and physical plausibility. Thus, the original initializations are expanded 10-fold by generating 3 variations with 3 stylization prompts. We refer the interested reader to App. B for more details.

**Metrics validation**   We use real-world videos as a reference baseline to validate that our evaluation metrics work as intended. By analyzing the metrics on these ground-truth recordings, we demonstrate the reliability of our approach and establish a reference performance level, representing the precision with which we measure adherence to physical laws. In essence, the metrics computed on real-world videos provide a baseline for how closely any generative model can align with physical principles.

To structurally analyze the videos, we extract the trajectories of the objects by applying promptable video segmentation. These trajectories comprise 2D coordinates of the recognized objects through time and are used for further analysis with our physical metrics (see Sec. 3.2 for details).

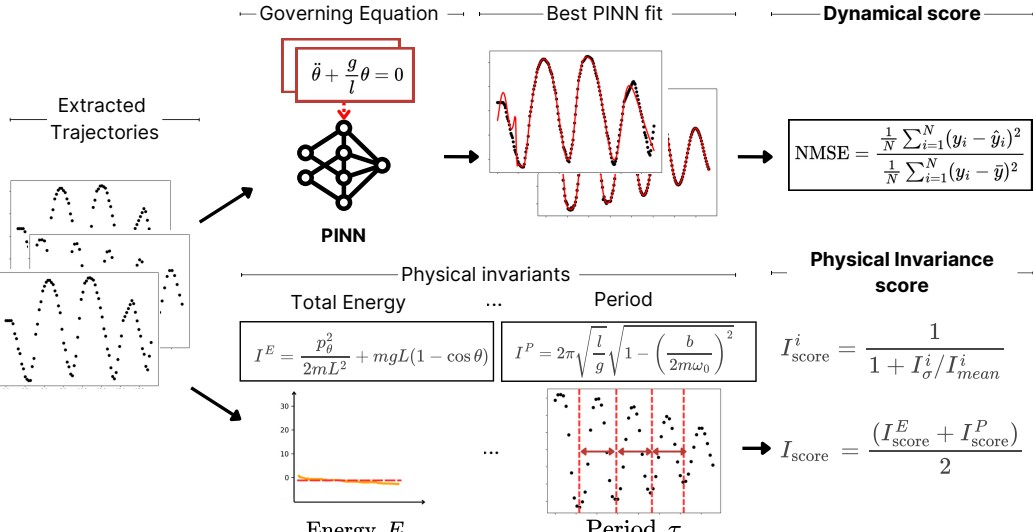

*Figure 4.* Evaluation of trajectories from real and VGMs videos using our Dynamical (upper) and Physical Invariance (lower) scores. For the Dynamical score, trajectories from real-world or generated videos are fitted to a PINNs with equations of motion as loss. For the Physical Invariance score, using the same trajectories, we estimate physical quantities that should be invariant in the respective systems, such as total energy and oscillation period, and use their variance as a measure of physical plausibility.

### 3.1. Prompting methods

The physical dynamics of a scene are determined by its initial conditions: object positions, velocities, and geometric constraints (e.g., shapes, rigid connections). In generative models, these conditions are set through prompting, which can take three forms: *a)* textual prompts, *b)* single-image prompts, and *c)* video (multi-frame) prompts, providing different levels of control over generation. *Textual prompts* offer only broad control, suggesting behaviors (e.g., rolling, falling) without precise states. *Single-image prompts* improve precision by fixing initial locations, but lack motion detail. Only *video prompts* specify both positions and velocities, providing the highest control.

We investigate how different levels of control affect the physical realism of generated samples. We explore both textual prompt enhancement and various multi-frame prompting for models capable of leveraging these features (e.g. (Agarwal et al., 2025; Yang et al., 2025)), allowing us to examine the relationship between input precision and output physical fidelity. Following (Yang et al., 2025), we use a VLM (Zeng et al., 2024) to expand simple scene descriptions to richer prompts via instruction templates in zero- or few-shot settings. As not all VGMs provide their own prompt upsampler, we rely on the ChatGLM family of models (Zeng et al., 2024), while for COSMOS-variants (Agarwal et al., 2025) and WAN-2.1 (Wang et al., 2025) we use their own devised (NVIDIA, 2024; Agrawal et al., 2024; Wang et al., 2024a) upsamplers respectively. In our evaluation, we create descriptive prompts with an emphasis on physical motion, and the upsampler brings the inference-time prompt distribution closer to that used during training.

### 3.2. Trajectory extraction

While generated videos could be directly evaluated in terms of 3D consistency (Liu et al., 2024b) or other pixel-level generation properties (Agarwal et al., 2025), such evaluations are limited to visual and geometric realism of the generated videos. Instead, we are interested in how well these videos conform to physical laws. This means that we need to extract the relevant physical state variables, such as the positions of objects, velocities, accelerations, masses, and other derived quantities. Thus, it is essential for further analysis to transform the generated videos into estimated state representations of the depicted objects and their trajectories.

As we need to track objects in both real-world and generated videos, with Segment Anything 2 (SAM-2) (Ravi et al., 2025) we reliably obtain 2D masks for any type of object in a zero-shot manner, using the Depth Anything (Yang et al., 2024a; Lin et al., 2026) screen for large depth variation. We annotate the first frame of our videos in the dataset with positive and negative labels. Given the masks generated by SAM-2 we extract the centroid of the object(s) (center of mass) in the video, at each frame of the video. In Appendix App. D.4 we provide a comprehensive robustness analysis showing that Morpheus is robust against tracking error accumulation.

For velocity, acceleration and angular velocity, we employ the *central difference method* (Swanson & Turkel, 1992). To further reduce noise, generated by the imperfections in the tracking pipeline, we follow with a series of smoothing operations, such as learning a linear regression with a

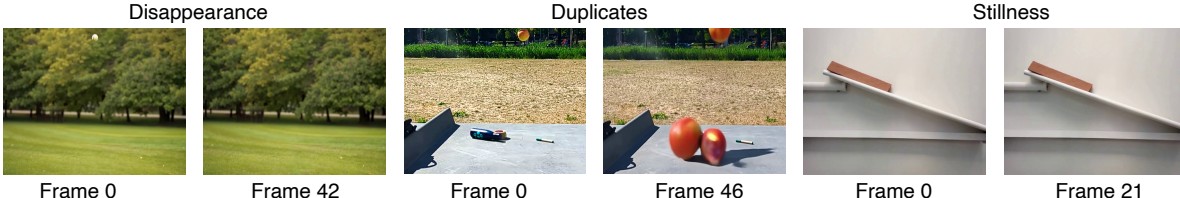

| Disappearance | | Duplicates | | Stillness | |
|---|---|---|---|---|---|
| Frame 0 | Frame 42 | Frame 0 | Frame 46 | Frame 0 | Frame 21 |

*Figure 5.* Different types of discarded generated videos: (left) A video showing the disappearance of the baseball ball during fall; (middle) A video illustrating of multiple apples falling; (right) A video in which the book should slide but it does not move at all.

sliding window and applying the Savitzky-Golay smoothing. The details can be found in App. C.

## 4. Physics-informed evaluation metrics

To assess the alignment of the generated video trajectories with physical laws, we propose a physics-informed evaluation framework to analyze physical experiments in both real-world and generated videos.

**Discard rate** As a first metric, we compute the *discard rate*, which reflects the proportion of model-generated samples that must be discarded to ensure reliable trajectory extraction needed for Physical Invariances and Dynamical scores. The discard filtering is automatic and consists of three criteria: First, we discard generated videos where objects lack sufficient permanence throughout the video. Second, we discard generated videos that do not have a consistent number of objects. Finally, we discard generated videos if there is little motion detected, as such videos are not suitable for physical analysis. The overall discard rate represents the proportion of generated videos that fail at least one of these criteria. In addition, we verify that none of the trajectories extracted from real-world are discarded, indicating that the filtering criteria do not remove physically valid real recordings. We provide further details on our filtering methodology in App. D.1. For the videos that pass filtering, we employ two metrics: *Dynamical score*, which measures adherence to the governing equation of motion, and *Physical Invariance score*, which quantifies invariance of conserved quantities such as energy or angular momentum (see Table 9 for all invariances).

**Physics beyond trajectory matching** Previous benchmarks such as PhysicsIQ (Motamed et al., 2026) and COS-MOS (Alhaija et al., 2025) use trajectory matching by comparing generated trajectories with ground-truth trajectories. As ground-truth trajectories themselves vary due to noise and hidden physical parameters that are not fully observable from visual image/video conditioning (e.g., object mass or friction), PhysicsIQ proposed to compare the obtained trajectory matching score with the variance in the real-world trajectories. However, such variance depends on the studied /recorded variations and can be arbitrarily large in cases

where hidden parameters vary significantly.

Trajectory matching operates at the pixel level and is therefore physically ambiguous. Even when multiple frames are used to constrain generation, a sampled trajectory may legitimately diverge from a single reference trajectory while still corresponding to a valid physical realization. For example, small differences in initial velocity, mass distribution, or unobserved forces can lead to distinct but physically consistent motions. Such cases are incorrectly penalized under trajectory-matching objectives, which implicitly assume a unique ground-truth trajectory. In contrast, **Morpheus** overcomes this limitation by evaluating generated videos at the level of governing physical laws rather than pixel-aligned trajectories from reference videos.

To make this concrete, we run a *real-vs-real* stress test: for each experiment we cross-compare two distinct real-world recordings of the *same* phenomenon (with slightly different, but equally valid, initial conditions) using the Physics-IQ (Motamed et al., 2026) trajectory-matching metrics. Since both videos are clean physical realizations, a metric that truly measured physical correctness should not penalize them. Instead, the overlap is low across the board (e.g., Spatiotemporal IoU of $0.010$ for projectile and $0.021$ for collision), confirming that minor unobservable differences make trajectory matching reject perfectly valid physics. The Per-variant numbers are reported in App. D.7.

### 4.1. Dynamical score

We calculate the dynamical score with physics-informed neural networks (PINNs) (Cuomo et al., 2022) that directly incorporate physical laws as a prior. This setting allows us to learn the physical trajectory that fits the data the most, without requiring explicit knowledge of initial conditions. Fig. 4 illustrates our approach. A PINN is a neural network that receives a timestep $i$ of the trajectory as input and outputs the trajectory coordinates $\hat{T}_i$, velocity $\dot{\hat{T}}_i$, and acceleration $\ddot{\hat{T}}_i$. The model is typically trained with a loss function, comprising two components $L_{\text{data}}$ and $L_{\text{physics}}$: $L_{\text{total}} = L_{\text{data}} + \lambda L_{\text{physics}}$, where the $L_{\text{data}}$ is responsible for fitting the model to the datapoints, $L_{\text{data}} = \frac{1}{N} \sum_{i=1}^{N} \|\hat{T}_i - T_i\|^2$, and $L_{\text{physics}}$ enforces physical law. For each experiment, the PINN loss function

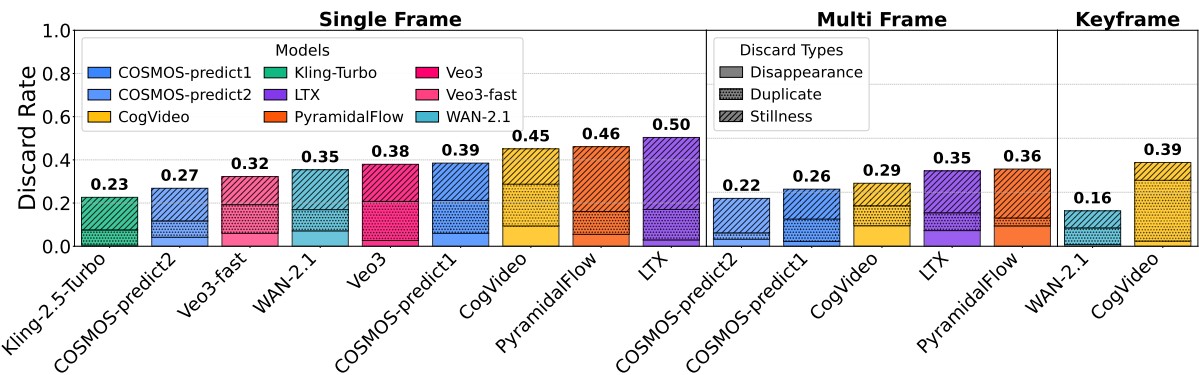

*Figure 6.* Average discard rates across all physical experiments (lower is better).

models the equation of motion as an ordinary differential equation, e.g., for a falling ball: $\dot{x} = 0; \quad \ddot{y} + g = 0$, where $y$ is the vertical position, $\ddot{y}$ is the acceleration and $g$ is the gravitational constant. The $L_{\text{physics}}$ is calculated as: $L_{\text{physics}} = \frac{1}{M} \sum_{j=1}^{M} \left\| \hat{\ddot{y}}_j + g \right\|^2 + \left\| \hat{\dot{x}} \right\|^2$, where $\hat{\ddot{y}}_j$ is the predicted acceleration from the PINN at the $j$-the time step.

**Computing the Dynamical score**   We use PINNs to assess the physical plausibility of trajectories from generated videos by computing the normalized mean square error (NMSE) of the model-learned trajectory derived from videos. We normalize by inverting the error, so 1 marks the best dynamical score and 0 a worse-than-constant PINN fit. As there is no explicit camera calibration that converts pixels to physical units, the absolute scale is unknown. PINNs, however, fit the functional form of the governing equations rather than absolute units. As a result, the Dynamical score is scale-invariant, as demonstrated in App. D.3. Higher Dynamical scores therefore correspond to higher physical plausibility. Full implementation details and hyperparameters are provided in App. D.5.

### 4.2. Physical Invariance score

To calculate a more fine-grained Physical Invariance score, we accompany each of our experiments with a list of physical invariances, i.e. values that we can derive from real-world trajectories that stay constant in time. As invariances vary per experiment, we present one case study for the falling ball experiments, and we include all other case studies in App. D.11 and Table 9.

**Case study– Falling ball**   physical invariants:

$\Rightarrow$ *Total energy.* Assuming negligible air resistance, the total energy — the sum of the kinetic and potential energy — of the ball is conserved. The kinetic energy of the ball is: $T = \frac{1}{2}m(v_x^2 + v_y^2)$, where $v = (v_x, v_y)$ is the speed of the ball and $m$ is it's mass. In addition, potential energy is

$V = mgy$ where $g$ is the gravitational acceleration constant and $y$ is the vertical coordinate. So, since the total energy is the sum of kinetic and potential, we get: $E = T + V = \frac{1}{2}m(v_x^2 + v_y^2) + mgy$.

$\Rightarrow$ *Energy-to-mass ratio.* Assuming that the mass of the ball is constant, we derive the following invariant: $\frac{E}{m} = \frac{1}{2}(v_x^2 + v_y^2) + gy = $ const, which we can estimate with the data from our trajectory.

$\Rightarrow$ *Acceleration;* since no external forces are acting on the ball except gravity, which is uniform and is directed downwards, the acceleration of the ball is also constant: $a_y = g = $ const.

$\Rightarrow$ *Horizontal momentum-to-mass ratio.* As with acceleration, the horizontal momentum, $p_x = mV_x$, is also preserved given no external forces.

**Computing the Physical Invariance score**   To convert the invariant into an actual score, like the Energy score, we calculate the standard deviation of the invariant time series and normalize it into the range of $(0, 1)$, with 1 indicating a perfect Physical Invariance score. As invariants must be by nature constant, a high standard deviation of these invariants (and thus a lower physical invariance score), indicates poor modeling of the respective physical invariants. In addition, for discarded trajectories we assign minimal Physical Invariance score equal to 0. A detailed score calculation procedure is described in App. D.12. For the derivations of each invariant we used, refer to App. D.11.

## 5. Experimental Analysis

We analyze the experimental results to understand not just *which* models perform best, but *why* they succeed or fail in capturing Newtonian dynamics. We present the aggregated scores for each model and conditioning type (single-frame, multi-frame, or keyframe-interpolation) in Fig. 7 and Fig. 6 respectively. Unless otherwise specified, we report the best scores obtained using either *enhanced* (an upsampled ver-

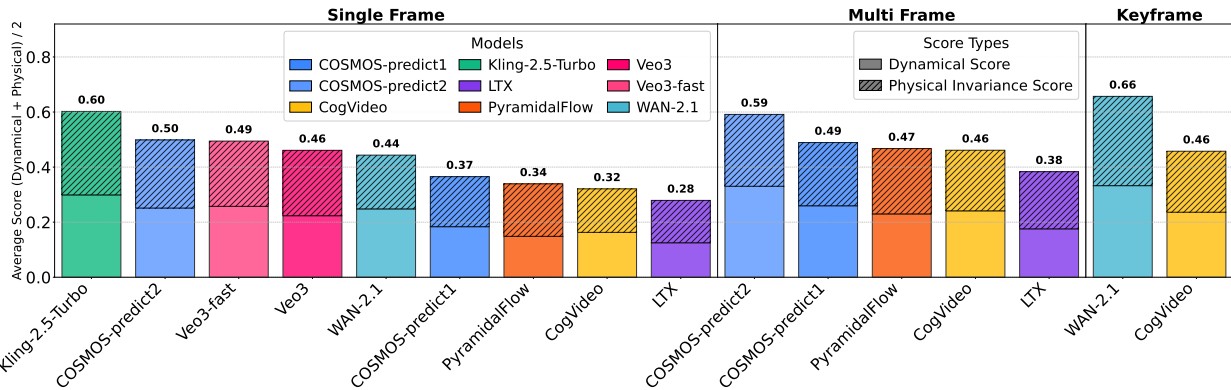

*Figure 7.* Aggregated scores across all physical experiments (higher is better).

sion of a *plain* prompt; the upsampling process is described in detail in Sec. 3.1) or *plain* (simple) textual prompts.

Real-world videos consistently serve as the upper bound for our evaluation, achieving high Dynamical Scores (0.96 − 0.99) and Physical Invariance Scores (≥ 0.90), without having any discarded samples; the per-experiment real-world scores are reported in Figure 8. In stark contrast, even the best performing models struggle to break past a combined score of 0.66 (Fig. 7), revealing a substantial gap between superficial *visual fidelity* and intrinsic *physical fidelity*. Per-experiment, per-metric breakdowns across all models are visualized as heatmaps in Figure 22 and Figure 21 (App. H).

**The Power of Temporal Context:** (i) *Keyframe-Interpolation :* Models conditioned on both the first and last frames (e.g., Wan2.1 Keyframe) dominate the benchmark, achieving the highest aggregate scores (0.66) and the lowest discard rates (0.16). This is consistent with the hypothesis that defining the *boundary conditions* of a physical event (start and end) significantly constrains the generation space, encouraging interpolation of a plausible trajectory rather than extrapolating into chaos. (ii) *Multi-Frame Extrapolation:* Providing a few consecutive frames as context (e.g., Cosmos-Predict2 Multi) improves performance over single-frame models (0.59 vs. 0.50), while also reducing discard rates by ∼ 5% . The temporal context helps the model to infer velocity and acceleration that are otherwise ambiguous in a static image. (iii) *Single-Frame Extrapolation* is the hardest setting. Even top-tier closed-source models like Kling-2.5 and Veo3 struggle with physical consistency when predicting the future from a single image, often resulting in high discard rates due to static or erratic motion.

**Model-Specific Insights** (i) *Open vs. Closed Source:* Powerful open-source models like Wan2.1 (with keyframe conditioning) and Cosmos-Predict2 are competitive and often outperform proprietary models like Veo3 family models and Kling-2.5. This suggests that physical reasoning capabilities are not solely a function of model scale or propri-

etary data, but may also depend on the architecture design choices, training dynamics, and conditioning strategies. (ii) *The "Stillness" Trap:* A major failure mode for models like PyramidalFlow and LTX-Video is to generate a video of an object staying still (see Fig. 26), yielding high discard rates (> 45%). (iii) *Hallucination & Duplication:* Closed source models, Veo3 and Kling-Turbo rarely produce static videos but suffer from object duplication or disappearance (Fig. 25) especially in the double pendulum case. This points to a different failure mode: while they model motion aggressively, they lose track of object permanence.

**Prompt Upsamplers** Improvements from enhanced prompts are highly *context*-dependent, see Fig. 28. Specifically, a significant boost performance is observed for Wan2.1 (+20.8% in multi-frame) and Cosmos-Predict2 (+27.9% in single-frame), likely by providing explicit language cues for guiding the expected dynamics. However, for LTX-Video and CogVideoX, enhanced prompts actually *affect* performance (up to −34%). This indicates that complex prompts mixing visual and physical instructions may confuse generations, at least in terms of physical invariants.

**Fine-Grained Physical Failures** Analyzing performance per experiment reveals more blind spots. Pendulum experiments are particularly challenging. While a closed-source model like Veo3 performs generally well, it collapses in pendulum generations compared to Veo3-fast and Wan2.1 (Fig. 29). This suggests that complex, constrained motion (rigid body rotation) is harder than free-falling motion (projectiles/falling). Models generally perform better on rolling/sliding tasks than on collisions or pendulums, likely because of richer training data in these settings. Also, rolling motion is visually smoother, less prone to chaotic behaviors, and less sensitive to precise conservation laws.

**Comparison with VLM-based evaluators** A natural question is whether a strong vision-language judge could replace our physics-informed scores. To test this, we evaluate our real-world recordings together with the generations con-

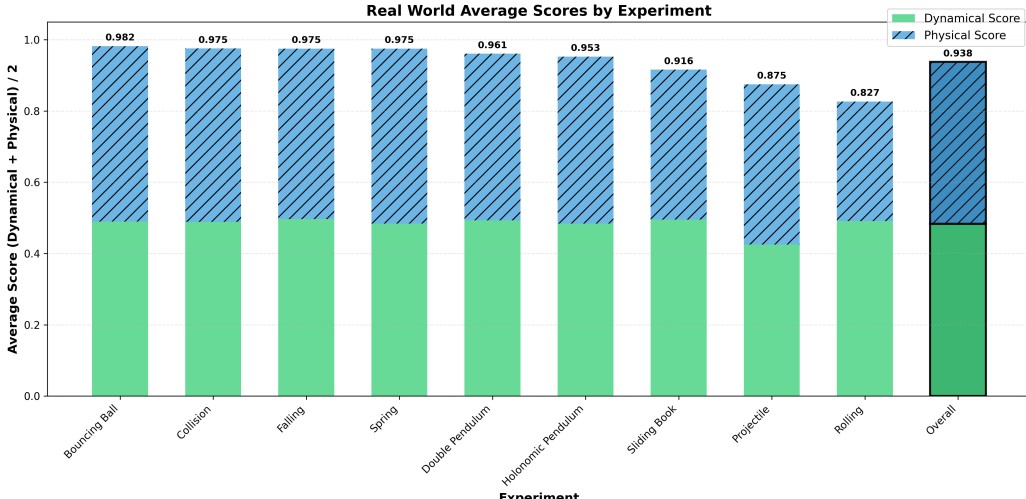

*Figure 8.* Morpheus scores for each experiment on real-world videos.

ditioned on them using VideoPHY2-AUTOEVAL (Bansal et al., 2026), a state-of-the-art VLM fine-tuned for physical commonsense, along its *Physical Commonsense* (PC, 1–5) and *Physical Rules* (PR, followed/violated) axes, and their strict intersection, the *Joint Physical Consistency* (JPC) pass rate (PC $\geq 4$ and PR $= 1$). A reliable evaluator should assign near-ceiling scores to our clean real-world recordings; instead it rates them only *moderately* (PC $= 3.71$, barely above the 3.53 it gives generations), shows no significant real-vs-generated gap in rule adherence (58.5% vs. 51.1%), and passes a mere 23.6% of real-world videos under JPC (only marginally over the 13.9% for generated). The VLM even hallucinates violations in clean physics while rewarding generations: e.g., it gives 0% JPC to real projectiles, bouncing balls, sliding books, collisions, and rolling cans, and assigns higher rule adherence to generated sliding books (82.8%) than to real ones (40.0%). Off-the-shelf VLM scoring is thus too brittle for precise physical evaluation; full per-model and per-category breakdowns are in App. D.6.

**Metric validation** We further validate that the **Morpheus** scores track human perception and are robust to trajectory-extraction noise. First, three judges rated 42 videos for physical plausibility (1–5); the total **Morpheus** score attains a **Spearman rank correlation of** 0.708 with the averaged human ratings (App. D.10). Second, we perturb extracted trajectories with Gaussian *jitter* ($\sigma = 0.5, 1.0$) and random *gaps* (up to 20% of points dropped): across all settings the total score deviates by $\leq 3\%$ in relative terms, with our smoothing block (see App. C) further reducing the deviation (App. D.8). The metric is likewise insensitive to the PINN architecture and the scoring-window length, with model rankings preserved (App. D.5, App. D.9).

In summary, the experimental analysis using **Morpheus**

reveals that even the latest video generation models generate samples that are visually appealing but are not consistent with the physical invariants of Newtonian mechanics.

## 6. Limitations

We limit our study to Newtonian mechanics because we focus on embodied AI settings, where rigid-body dynamics form a foundational layer of physical interaction, while also making physics-informed neural network optimization more reliable when operating on noisy video trajectories. Our evaluation focuses on short videos with no camera motion to reduce confounding factors in trajectory extraction and ensure consistency across state-of-the-art video generative models. That said, the underlying principles of physics-informed metrics are agnostic to video length, and can be used when longer-horizon allows for consistent generations.

## 7. Conclusion

**Morpheus** studies how well modern video generation methods model Newtonian mechanics of rigid-body systems in terms of physical invariants and compliance with Newtonian physical constraints. This is a necessary step toward aligning video generation with robot world models and embodied AI applications, where adherence to physical laws is critical for reliable decision-making. We construct physics-informed evaluation metrics that model precisely the physical invariants of various real world physical systems of Newtonian mechanics, and evaluate state-of-the-art video generation with advanced prompting techniques. Results show that video generation models improve significantly over time in terms of visual realism, however, even the best of them are far from encoding Newtonian mechanics accurately enough to claim physical world model capabilities.

## Impact Statement

This paper presents work whose goal is to advance the field of Machine Learning. We believe that the consequence of work have no negative ethical or societal impact. This work involves the collection of a real world recording and generated samples and the evaluation of publicly available models, all used in compliance with their original licenses. No new human data or personal annotations were gathered. Our study does not any collection or processing of user data.

## Acknowledgments

Antonios Tragoudaras gratefully acknowledges ELLIS Unit Amsterdam for the travel grant, financially supporting the presentation of this work at the Forty-Third International Conference on Machine Learning (ICML 2026).

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

# APPENDIX

## A. Dataset

Overall, we conducted a set of 9 core physical experiments, highlighting different physical principles, including:

1. Falling: Objects dropped from rest until they make impact with the surface, used to test uniform gravitational acceleration and energy conservation.

2. Projectile motion: A ball launched at various initial velocities and angles, testing the preservation of momentum and energy, as well as the uniformity of gravity.

3. Rolling: A metal can rolling from a slope, with energy conservation.

4. Sliding: A book sliding from a slope, with energy conservation.

5. Holonomic pendulum: A ball affixed to a rigid rod, with periodic motion and energy conservation.

6. Double pendulum: A more complex system with a pendulum attached to another pendulum, illustrating chaotic behavior and conservation laws in nonlinear dynamics.

7. Bouncing: A ball observed from the moment it first impacts the surface until it rebounds and impacts again, testing gravitational acceleration and energy conservation in a more challenging setting.

8. Collision: Two metal balls with the same or different masses collide with each other. One of them is initially stationary and the other one collides with it.

9. Spring: A weight hanging below a vertical spring, perform up and down simple harmonic vibration with energy conservation.

For each system, we recorded multiple times the type of experiment trying to have homogenous videos, while after a few iterations, we varied the initial conditions or configuration parameters. Table 1 below summarizes the number of recordings and configurations for each experiment.

| Experiment | Videos | Factors of Variation | Configuration Initial Condition Description |
|---|---|---|---|
| Falling | 20 | 2 | Object type and height from which the object was released. |
| Projectile | 15 | 3 | Angle of launch, slingback extension levels, launched ball color. |
| Bouncing ball | 12 | 1 | Heights from which the ball was released before bouncing. |
| Holonomic Pendulum | 22 | 1 | Initial angle from the vertical (zero-degree resting position). |
| Double Pendulum | 10 | 1 | Initial height of the second (top) pendulum bob. |
| Rolling | 15 | 2 | Incline angle of the ramp from which the object was released and object type. |
| Sliding | 10 | 1 | Incline angle of the ramp (slope) from which the object was released. |
| Collision | 12 | 3 | Masses of the colliding objects and their initial velocities before impact. |
| Spring | 7 | 1 | Magnitude of the initial force/impulse used to displace the mass from rest. |

*Table 1.* Summary of the experimental dataset from real-world recorded videos.

For each experiment, we provide representative frames: falling objects in Figure 9; bouncing, projectile motion, holonomic pendulum, and double pendulum in Figure 10; rolling in Figure 11; and sliding, collision, and spring in Figure 12.

**Falling**   For the falling experiment, we started with a standard table tennis orange ball as the simplest object. A mechanical actuator [1] was used to hold the ball in place at a certain height (initial position) and as a release mechanism to control the moment the ball fell free, before making contact with the surface below. Different height levels from the surface were used as initial positions, resulting in trajectories with different lengths (smaller or larger). In addition to the ball, we conducted extra experiments using different everyday objects, namely a plastic whiteboard marker, an adhesive tape roll, and an apple.

---

[1]*Motor Model: T825, Motor serial number: 00362129*

Unlike actuator controlled experiments, these objects were released directly from the hand of a person at varying initial heights. This setup introduced additional variability in the conditions and the orientation of the object.

**Bouncing ball** The bouncing ball experiment begins immediately after the falling ball makes impact with the surface. It focuses on observing the ball during its bounce, capturing its trajectory as it rebounds upwards after contact with the surface.

**Projectile** For this experiment, a custom 3D printed projectile was built, along with three different balls of the same plastic material but of different colors. The projectile works with string rubber bands following the same principle of a slingback. During our recordings, we varied three different parameters. The angle of the launch for the ball, the force with which the ball was launched into the air, and the color of the ball.

**Holonomic pendulum** For this setting, a rigid metal structure consisting of a pole, perpendicular to the ground, on which a solid metal stick was mounted. The joint holding the stick was adjusted to allow for a normal friction coefficient, resulting in an intuitive retrogressive back-and-forth movement simulating a typical pendulum oscillatory trajectory. At the end of the metal stick a small table tennis ball was attached, as the SAM2 predictor can confidently track the center of the ball aligning with the central axis at the end of the stick. Using the zero angle as the resting position, we varied the angle at which the pendulum was released resulting in distinct retrogressive trajectories. As in the falling ball experiment the same release mechanism model was employed to manipulate the moment the pendulum was let freely to swing.

**Double pendulum** A custom structure consisting of a wooden base, a metal pole mounted on the top of the base, and a joint mounted at a degrees angle to the center axis of the pole, to keep the longer bob of the pendulum in place. These structures ensure that each 3D printed plastic bobs of the pendulum can rotate freely with normal friction resulting in the typical chaotic motion double pendulum are known for. A double pendulum consists of two bobs attached end-to-end. Each pendulum has its angle relative to the vertical. The same release mechanism as in previous experiments is utilized to define the starting position of each pendulum link. This starting position can be described as the angle each bob makes with the vertical when it is still stationary.

**Rolling** For this setting, we examined objects that roll down an inclined surface. We used three different objects: a full can, which was sealed, an empty can, and an orange. The slope of the surface was adjustable, allowing us to vary the steepness of the slope. The objects were placed by hand at the top of the slope before being released. Due to different mass distributions and shapes, we had different rolling behaviors across these three objects.

**Sliding** For this experiment, we investigated the sliding motion on an inclined surface using a flat book. We varied the slope of the surface between trials. The book was placed by hand at the top of the slope before being released. Depending on the angle of the slope, the trajectories exhibited smooth sliding motion.

**Collision** In the collision experiment, we studied collisions between objects of different sizes. Three setups were recoded: A large object collides with a smaller one, two objects of equal size collide with each other, and a small object collides with a larger one. For all cases, the initial velocity of the moving object (leftmost object with the right one at rest) was introduced with a gentle push at random speeds. These settings produced a diverse outcome for the aforementioned experiment, depending on the relative mass of the objects and the initial velocity.

**Spring** In this spring experiment, we analyzed the oscillatory motion of a cylindrical metal weight suspended from a vertical spring. The object was initially displaced from its rest position by hand with a small (but random) force and then released. The amplitude of oscillation was defined by this displacement and the restoring force led the object to move in a vertical periodic movement until equilibrium. Initial force was the main reason for the motion, with no actuator being employed, and gradually over time decayed due to damping effects.

## B. Augmentation with Cosmos-Transfer1

We augment a subset of experiments with style transfer to diversify object appearances while preserving underlying physical dynamics, creating initializations more representative of typical VGM training data. Specifically, we apply COSMOS-Transfer to five experiment types—Falling ball, Projectile, Holonomic pendulum, Rolling, and Sliding. For each experiment type, we select representative videos and generate style-transferred variants that alter object semantics and appearance

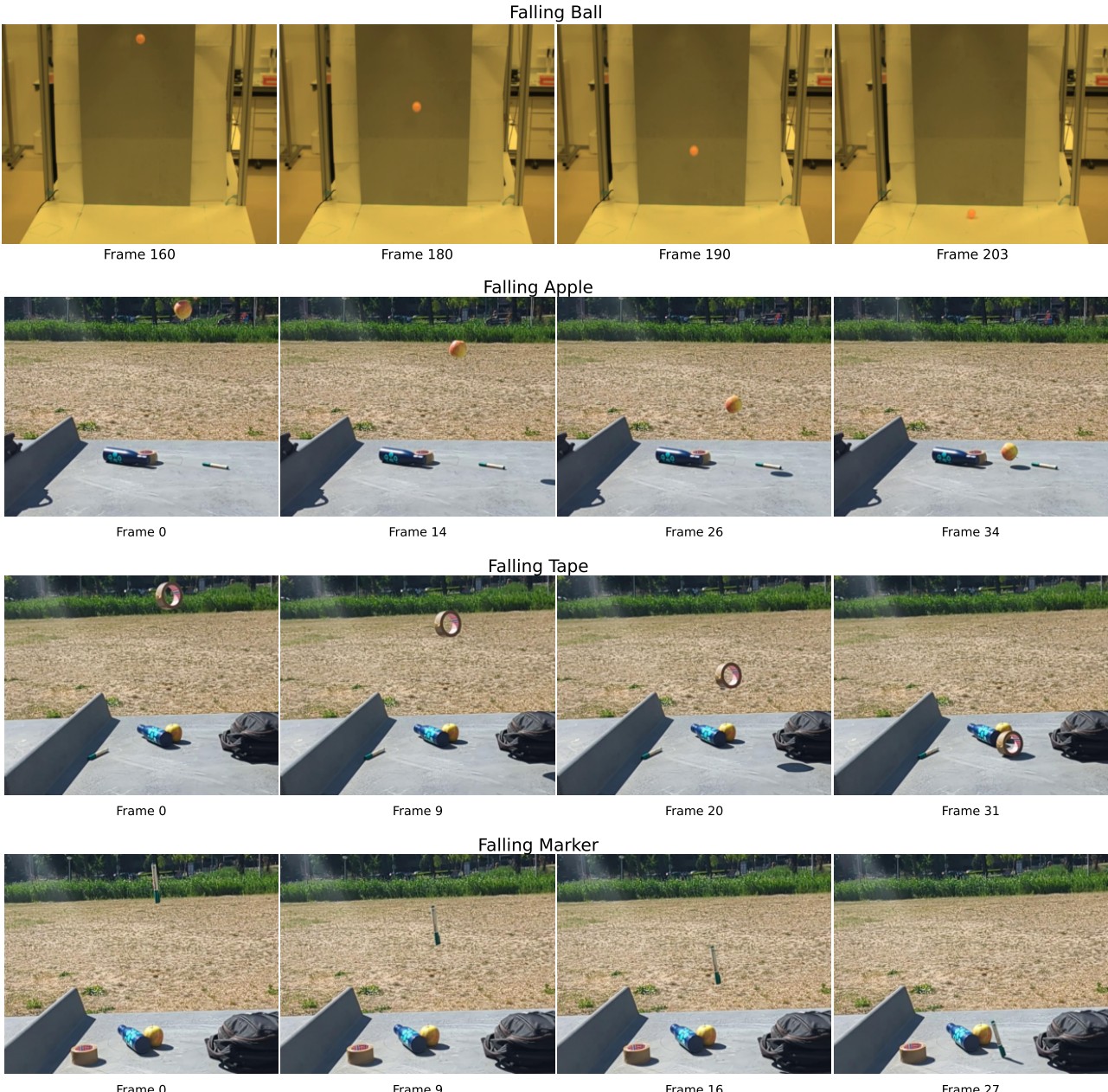

*Figure 9.* Representative frames from the falling experiments in the **Morpheus** benchmark: ball, apple, tape, and marker.

while retaining motion consistency. This yields three transferred variants per original video (two for Holonomic pendulum), providing diverse yet physically plausible conditioning scenarios. Per experiment type, this produces $3 \times 3 = 9$ augmented conditioning scenarios for most experiments, and $3 \times 2 = 6$ for Holonomic pendulum.

The transfer process uses per-frame object masks from SAM2 (Ravi et al., 2025), reusing the same masks from our trajectory-extraction pipeline to isolate only the object under study. We provide concise, object-focused prompts describing the desired replacement. All transfers undergo manual screening for visual and physical plausibility. When artifacts such as camera motion, temporal drift, hallucinations, shape misalignment, or mask leakage are observed, we enable Canny-edge guidance in COSMOS-Transfer with an edge-conditioning weight of 0.5, with the object mask now accounting for the other 0.5 conditioning weight. This additional constraint preserves scene structure while still allowing complete style transformation.

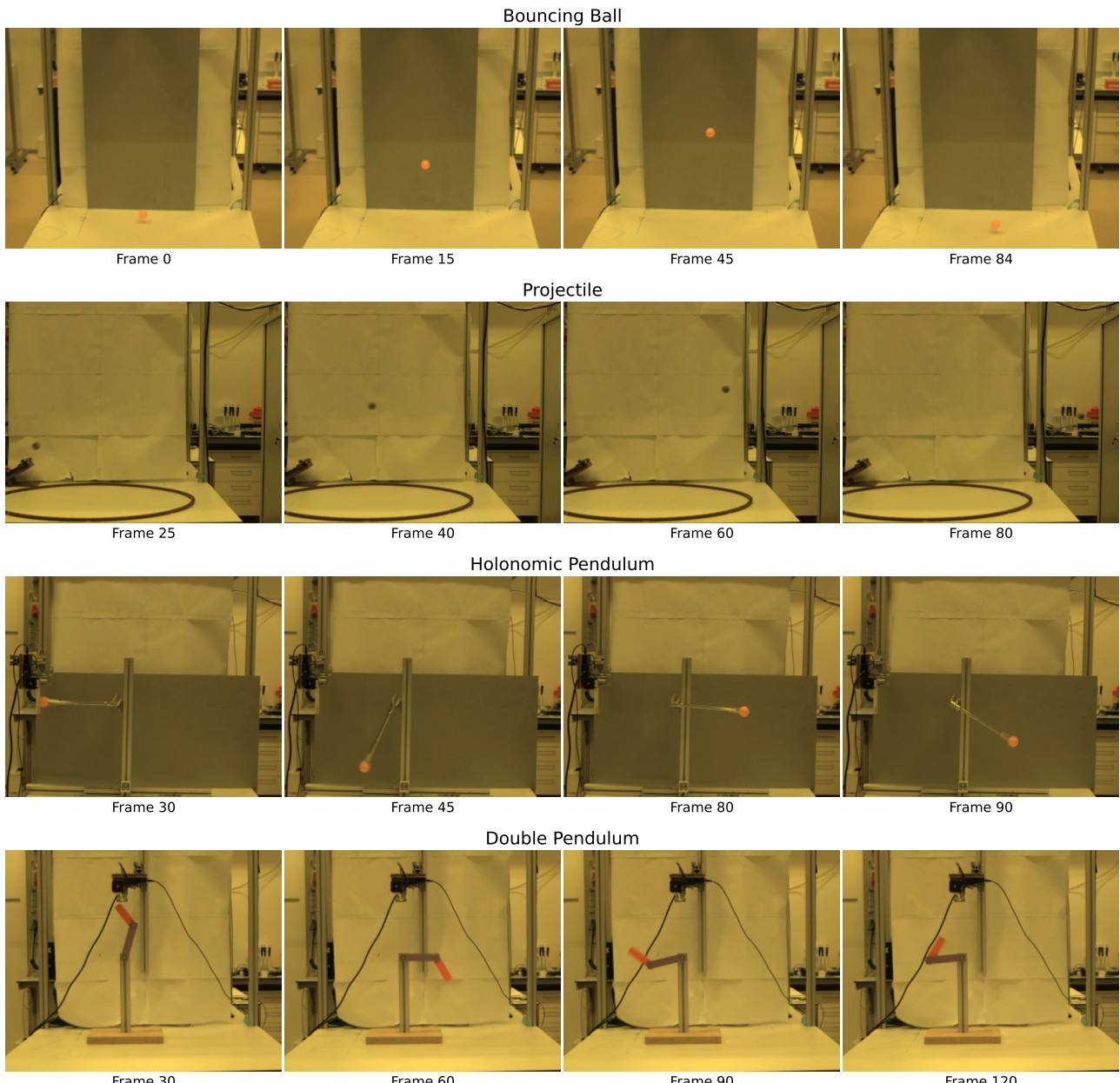

*Figure 10.* Representative frames from four experiments in the **Morpheus** benchmark: bouncing ball, projectile motion, holonomic pendulum, and double pendulum.

Figure 13 shows representative examples comparing the first frames of original videos with their style-transferred variants. These augmentations effectively multiply our conditioning scenarios: each original video contributes multiple physically consistent variants, expanding the diversity of initial conditions for VGM evaluation.

## C. Velocity and acceleration estimation

We estimate objects' velocity and acceleration from the extracted trajectory using multiple stages.

We use the central difference method for most points in the time series. This method computes velocity by considering both

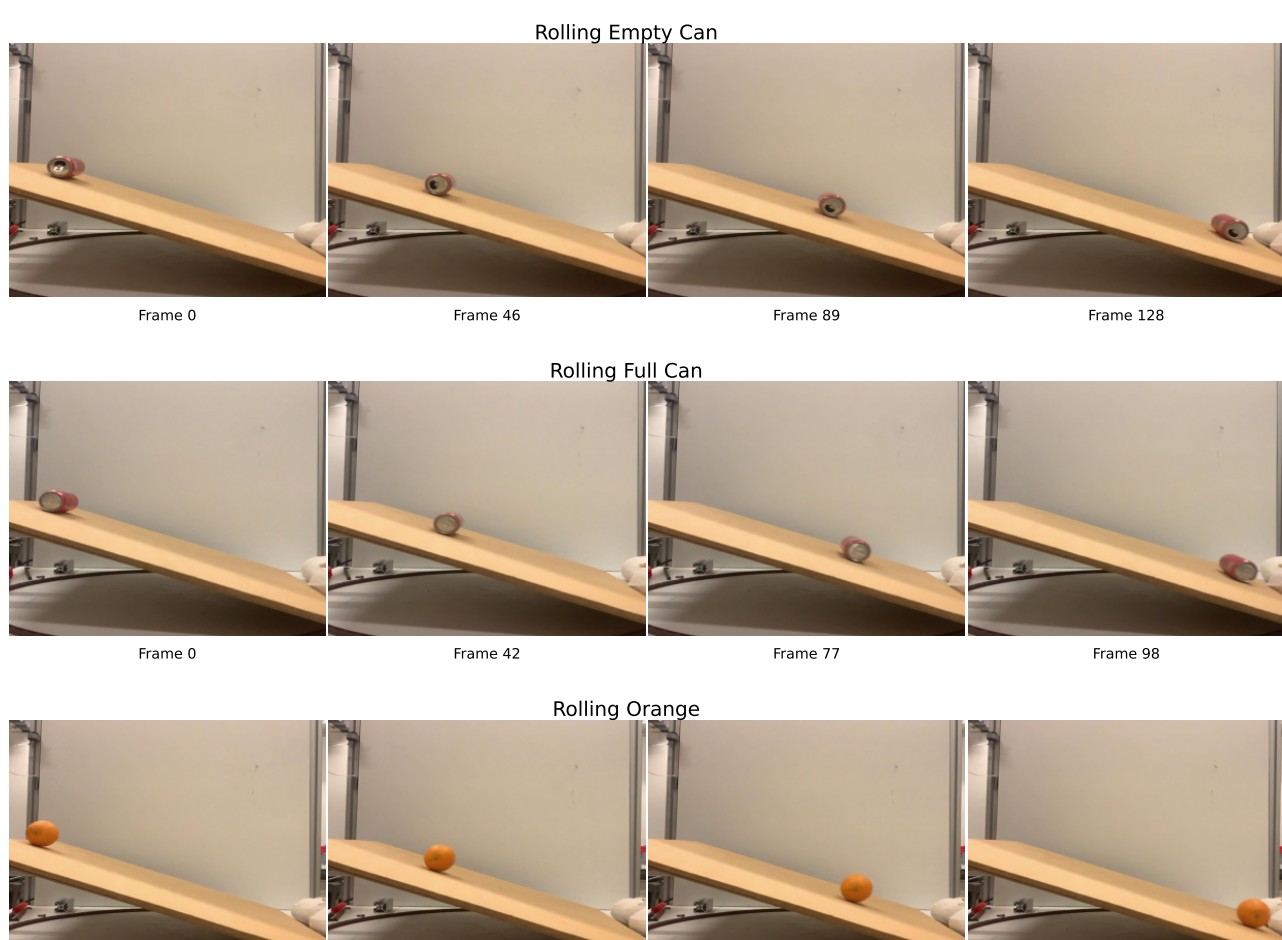

*Figure 11.* Representative frames of the rolling experiments in the **Morpheus** benchmark: an empty can, a full can, and an orange, each released by hand on an inclined surface of varying slope.

forward and backward positions, reducing single-sided differentiation errors.

$$v_i = \frac{x_{i+1} - x_{i-1}}{t_{i+1} - t_{i-1}}, \quad 1 \le i \le N - 2 \tag{1}$$

Since the central difference is not applicable at endpoints, we use one-sided differences. Forward difference (starting point):

$$v_0 = \frac{x_1 - x_0}{t_1 - t_0}$$

Backward difference (ending point):

$$v_N = \frac{x_N - x_{N-1}}{t_N - t_{N-1}}$$

To enhance precision, we perform linear regression within a sliding window.

$$x(t) = vt + b \tag{2}$$

The velocity (slope) is solved using the least squares method with window size w:

$$\begin{bmatrix} v \\ b \end{bmatrix} = \left(A^T A\right)^{-1} A^T x \tag{3}$$

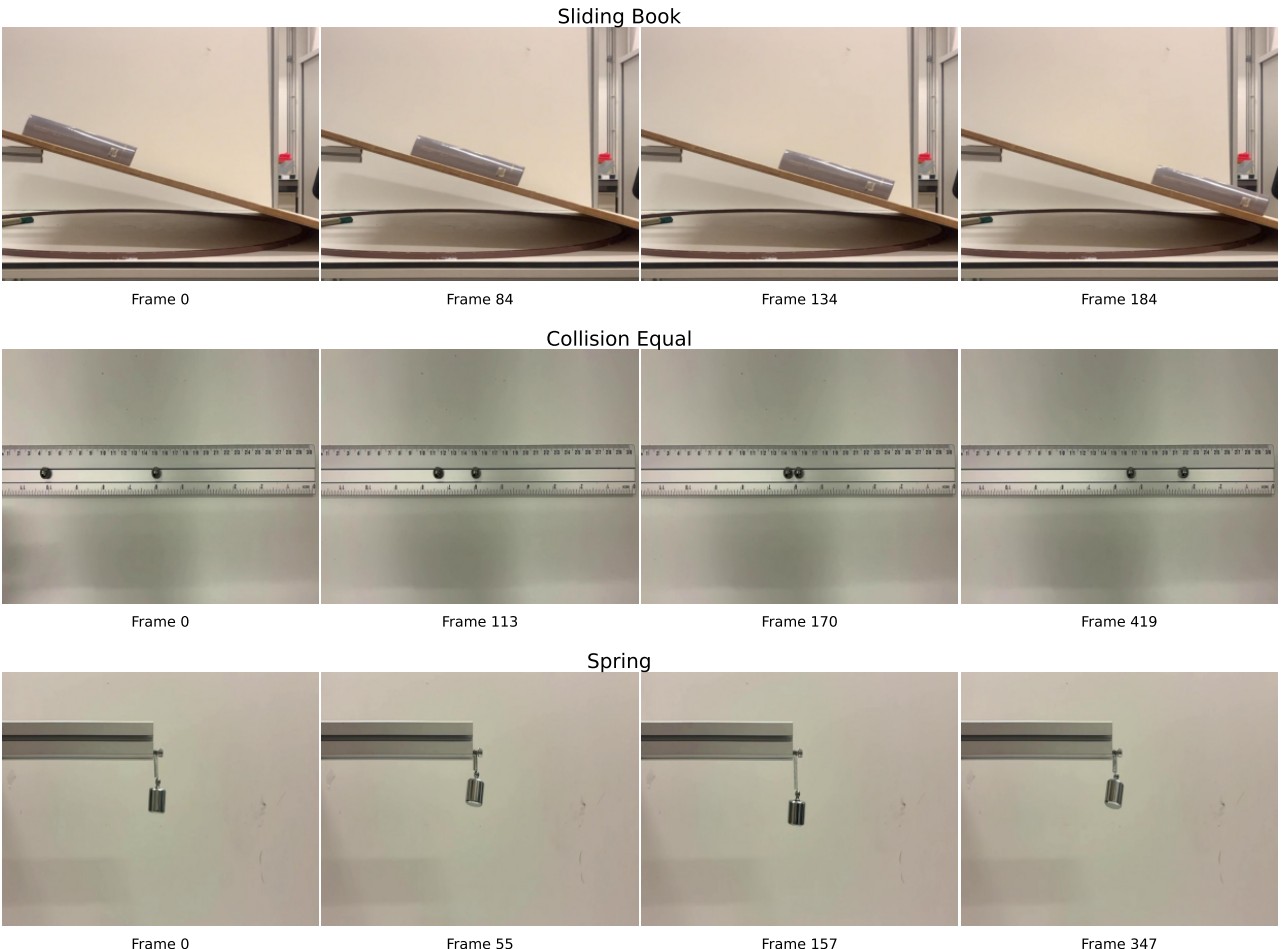

*Figure 12.* Representative frames from three experiments in the **Morpheus** benchmark: sliding, collision (equal-sized objects), and spring.

where matrix A contains time information.

$$A = \begin{bmatrix} t_1 & 1 \\ t_2 & 1 \\ \vdots & \vdots \\ t_w & 1 \end{bmatrix} \tag{4}$$

We combine linear regression and central difference results using weighted averages.

$$v_{\text{final}} = \alpha v_{\text{regression}} + (1 - \alpha) v_{\text{central}} \tag{5}$$

Here $\alpha = 0.7$, indicating greater confidence in the regression method. Finally, we apply Savitzky-Golay filtering for smoothing (Luo et al., 2005). This step effectively removes high-frequency noise from velocity calculations.

$$v_{\text{smoothed}} = \text{SG}(v_{\text{final}}, \text{window}, 3) \tag{6}$$

The entire calculation process can be summarized as:

$$v(t) = \text{SG}\left(\alpha v_{\text{regression}}(t) + (1 - \alpha) v_{\text{central}}(t), w, 3\right) \tag{7}$$

where w is the window size (odd number for symmetry); $\alpha = 0.7$ is the weighting coefficient; SG represents Savitzky-Golay filter of order 3; Regression window range: $[t - w/2, t + w/2]$.

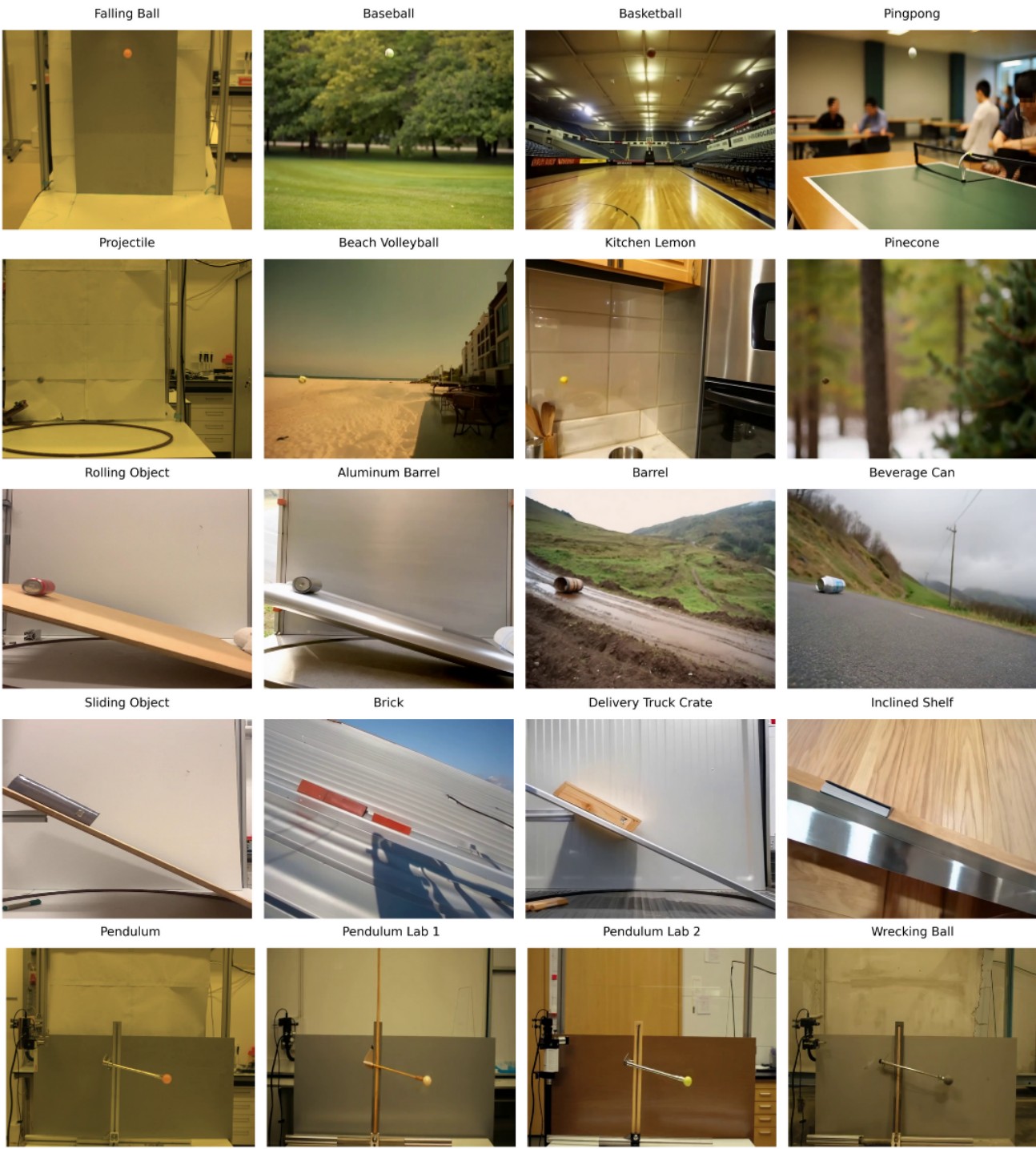

*Figure 13.* COSMOS-Transfer style augmentations for physics experiments. Each row displays the original experiment (left) with its style-transferred variants (right), showing how object appearance can be altered while preserving physical dynamics. The transfers provide diverse yet physically consistent initializations for VGM evaluation across five fundamental physics scenarios.

For the acceleration, we first calculate the acceleration using the central difference. For $1 \leq i \leq N - 2$:

$$a_i = \frac{v_{i+1} - v_{i-1}}{t_{i+1} - t_{i-1}} \tag{8}$$

Dealing with the endpoints using the same metric as velocities, we get the final acceleration for the entire trajectory.

$$a_0 = \frac{a_1 - a_0}{t_1 - t_0}$$
$$a_N = \frac{v_N - v_{N-1}}{t_N - t_{N-1}}$$

# D. Evaluation metrics

| Benchmark | Real-world ground-truth | Quantitative evaluation | Initial condition grounding | Physical laws evaluation |
|---|---|---|---|---|
| VideoCon-Physics | ✓ | ✗ | ✗ (only text) | ✗ |
| VAMP | ✓ | ✓ | ✗ (only text) | ✗ |
| PhyGenBench | ✓ | ✗ | ✗ (only text) | ✗ |
| Kang et al. (2025) | ✗ | ✓ | ✓ (1 or 3 frames) | ✗ |
| COSMOS | ✗ | ✓ | ✓ (image + video) | ✗ |
| Physics-IQ | ✓ | ✓ | ✓ (image + video) | ✗ |
| **Morpheus** (ours) | ✓ | ✓ | ✓ (image + video) | ✓ |

*Table 2.* Comparison of physics-based video understanding benchmarks. Our benchmark is the first to use real physical laws for evaluation. Symbols: ✓= supported, ✗= not supported

## D.1. Discard rate

We generate $N_{total}$ videos for each type of experiment. Among these videos, we discard those that do not meet our quality standards, following a three-stage filtering out. First, we discard videos where object are disappearing from the videos the number of such videos is $N_{disappear}$. Second, we analyze the number of objects in each video and discard videos that do not maintain a consistent object count in the not discarded yet videos. For this purpose we employ DEVA tracking (Cheng et al., 2023) built on top of Grounded SAM (Ren et al., 2024) (with object names from the prompt as Grounding DINO (Liu et al., 2024a) query) for consistent open-vocabulary prediction of 2D object masks. We denote the number of discarded videos in this step as $N_{duplicate}$. Specifically, we evaluate the proportion of frames containing multiple objects. Videos are filtered out if this proportion exceeds a predetermined threshold. Finally, we discard videos where the motion is too small to be meaningful in the not discarded yet videos, the number of such videos is $N_{stil}$. The overall *discard rate DR* is defined as

$$DR = \frac{N_{disappear} + N_{duplicate} + N_{still}}{N_{total}}.$$

For redundancy we manually inspect all filtered choices by DEVA tracking to make sure we do not yield false positive or negatives respectively.

## D.2. Robustness to non Ideal Effects

While Morpheus evaluates against ideal Newtonian laws (e.g., conservation of energy), real-world experiments inevitably include dissipative forces like air resistance and friction. A key validation of our benchmark is that real-world videos still achieve extremely high scores. This empirically demonstrates that for the objects and velocities in our dataset, these non-ideal effects are negligible relative to the gross violations exhibited by current video generation models. Thus, our framework does not unfairly penalize plausible, minor dissipative effects; rather, it flags the massive deviations (e.g., erratic acceleration) typical of current generative models.

## D.3. Scale Invariance Verification

To confirm that our Dynamical evaluation pipeline is insensitive to precise camera calibration, we conducted a scaling experiment where each real world video trajectory was rescaled between a range of scaling factors (0.50, 0.75, 1.00, 1.25, 1.50, 1.75 and 2.00). Throughout this entire range the Dynamical score stays extremely close to 1, with minimal variations. The results for the pendulum, free fall and sliding experiments can be visualized in Figure 14. This clearly indicates that the PINN is capable of recovering the underlying physical dynamics when the absolute pixel changes within a reasonable range. In practice, this would mean that small differences in camera distance or object size do not affect the Dynamical Score. Therefore, no explicit camera calibration is needed for Morpheus and its evaluation is robust under natural scale variability in the real world setting.

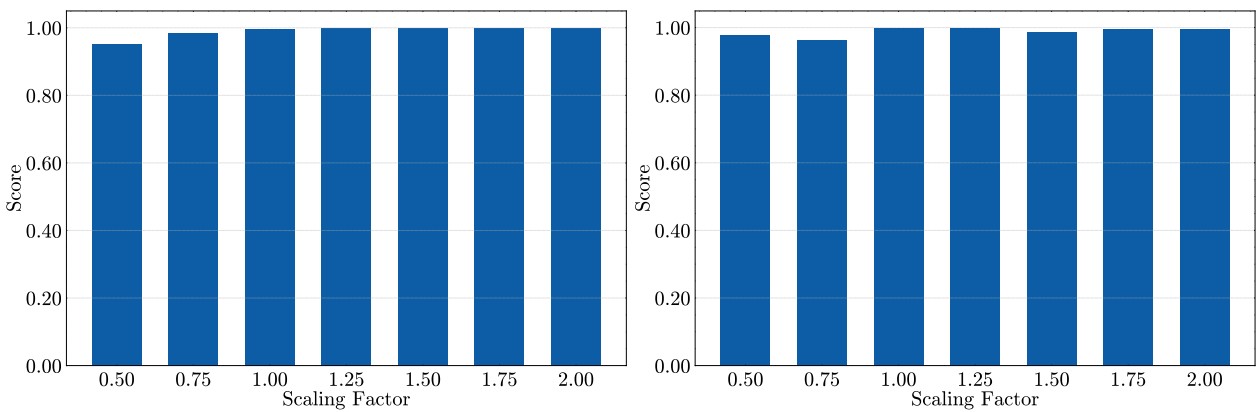

*Figure 14.* Scale invariance for Freefall (left) and Pendulum (right).

## D.4. Depth Consistency Evaluation

A core concern in automated benchmarking is whether the evaluation pipeline itself introduces errors that could be mistaken for model failures. In all the studied experiments, the video camera is orthogonal to the object's motion and is fixed. This allows us to compute the Physical Invariance and Dynamical scores using only information extracted from 2D pixel space, available for generated videos. The results are presented in Fig. 16, showing that most of the models are reasonably consistent and thus object properties like energy conservation could be also studied using only 2D coordinates.

To ensure that real-world and generated videos meet the static orthogonal camera assumption necessary to assess 2D trajectories as a projection of 3D motions, we incorporate a depth consistency check using Depth Anything V3 (Lin et al., 2026). We calculate per-frame depth estimates and check that depth is roughly constant over time. This process verifies that the objects do not perform any unnatural motion along the camera's optical axis, which otherwise would invalidate physical measurements of velocity, acceleration or energy conservation computed only from the 2D pixel coordinates. Our metrics (Dynamical Score and Physical Invariance Score) depend on accurate 2D trajectories. For that reason the above depth consistency check is an important quality control step. This demonstrates that real-world recordings meet the static camera prerequisite. In addition, we also evaluate the same check with the newest Depth Anything V3, which enhances spatial sharpness and temporal stability. As illustrated in Fig. 15 DA2 and DA3 exhibit high depth stability across all physical experiments, with an average depth consistency scores of 98.3% and 96.8%, respectively. Each individual experiment exceeds 96% consistency for both models, which confirms that the objects are at constant distance from the camera throughout the motion. This confirms that all trajectories lie within a near planar imaging regime, justifying our use of 2D trajectories for physical evaluation. The strong agreement between DA2 and DA3 establishes that our conclusions are robust to this specific choice of depth estimators.

## D.5. Physically-informed Neural Networks

Unlike typical neural networks, which are normally trained only on data, prior knowledge about the physical system is integrated into PINNs. This prior knowledge of the physical system, often in governing physical laws such as Newtonian mechanics or energy conservation, is imposed during training. Given that the system modeled from the generated videos is known from the provided prompt, the training process incorporates these laws into the loss function. The total loss for a PINN is defined as:

$$L_{\text{total}} = L_{\text{data}} + \lambda L_{\text{physics}}, \tag{9}$$

where $L_{\text{data}}$ ensures that the network's output can match the observed data. At the same time, $L_{\text{physics}}$ is penalizing deviations from the governing physical equation and $\lambda$ is a hyperparameter balancing the contribution of the each loss component. In this way, PINNs can bring both data and physical laws together during training while being consistent with the underlying physical system. For a trajectory $T$, the data loss is defined as

$$L_{\text{data}} = \frac{1}{N} \sum_{i=1}^{N} \|\hat{T}_i - T_i\|^2, \tag{10}$$

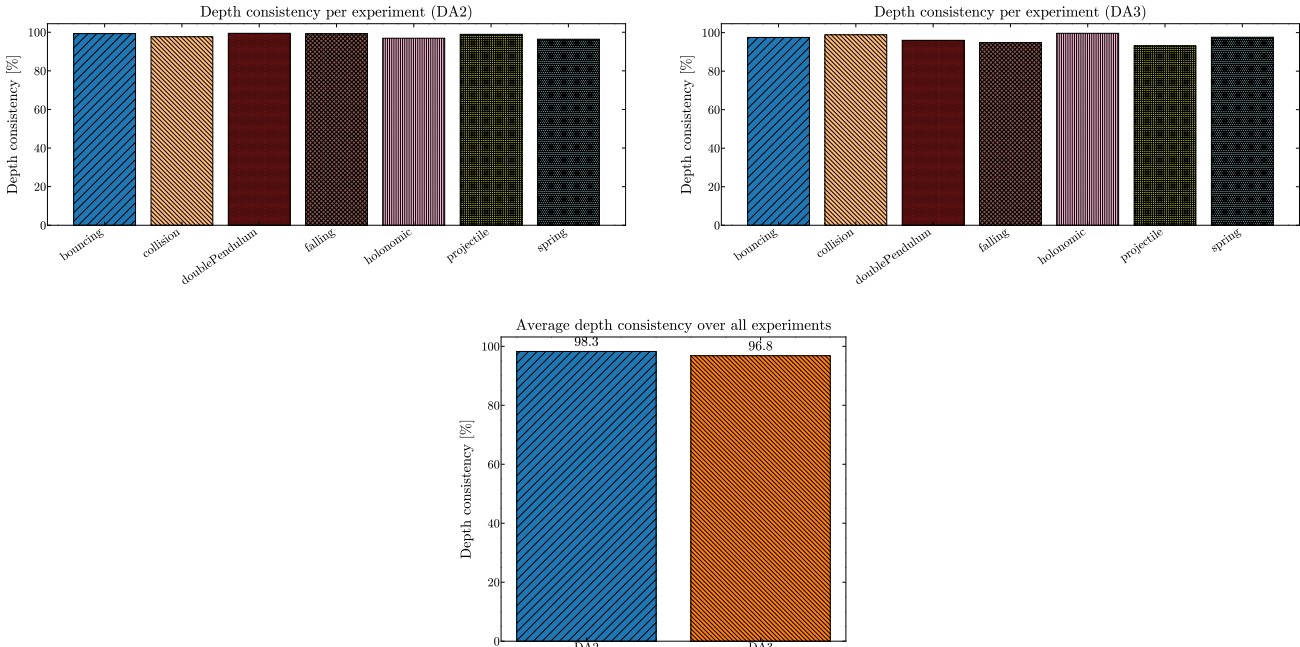

*Figure 15.* Depth consistency across seven real-world Newtonian experiments for Depth Anything V2 (left) and V3 (right). Bars report per experiment depth consistency, while the bottom figure summarizes the average over all experiments, showing that both models produce highly stable depth estimates (98.3% for DA2 vs. 96.8% for DA3).

, where $\hat{T}_i$ is the trajectory predicted by the network at the $i$-th timestep and $T_i$ is the corresponding ground truth trajectory at the same timestep. On the other hand, the physics loss is derived separately for each experiment, given the nature of the system's dynamics. The motion of a free-falling object follows:

$$\ddot{y} + g = 0, \tag{11}$$

where $y$ is the vertical position, $\ddot{y}$ is the acceleration and $g$ is the gravitational constant. This means that for this phenomenon, the loss is defined as: The physics loss for free fall is defined as:

$$L_{\text{physics}} = \frac{1}{M} \sum_{j=1}^{M} \left\| \hat{\ddot{y}}_j + g \right\|^2, \tag{12}$$

where $\hat{\ddot{y}}_j$ is the predicted acceleration derived from the PINN at the $j$-th time step. The motion of a holonomic pendulum is governed by:

$$\ddot{\theta} + \frac{g}{l} \sin \theta = 0, \tag{13}$$

where $\theta$ is the angular displacement, $l$ is the pendulum length and $g$ is the gravitational constant.

The corresponding physics loss is:

$$L_{\text{physics}} = \frac{1}{M} \sum_{j=1}^{M} \left\| \hat{\ddot{\theta}}_j + \frac{g}{l} \sin(\hat{\theta}_j) \right\|^2, \tag{14}$$

where $\hat{\ddot{\theta}}_j$ and $\hat{\theta}_j$ are the network-predicted angular acceleration and displacement, respectively, at the $j$-th timestep. In the present work, we use the Dynamical score to evaluate how well the predicted trajectories align with the ground truth. The Dynamical score is derived from the Normalized Mean Squared Error (NMSE), which provides a relative measure of error by normalizing the Mean Squared Error (MSE) with the variance of the ground truth trajectory. Main motivation behind this choice, is to make the evaluation independent of scale. The NMSE is calculated as:

$$\text{NMSE} = \frac{\frac{1}{N} \sum_{i=1}^{N} (y_i - \hat{y}_i)^2}{\frac{1}{N} \sum_{i=1}^{N} (y_i - \bar{y})^2}, \tag{15}$$

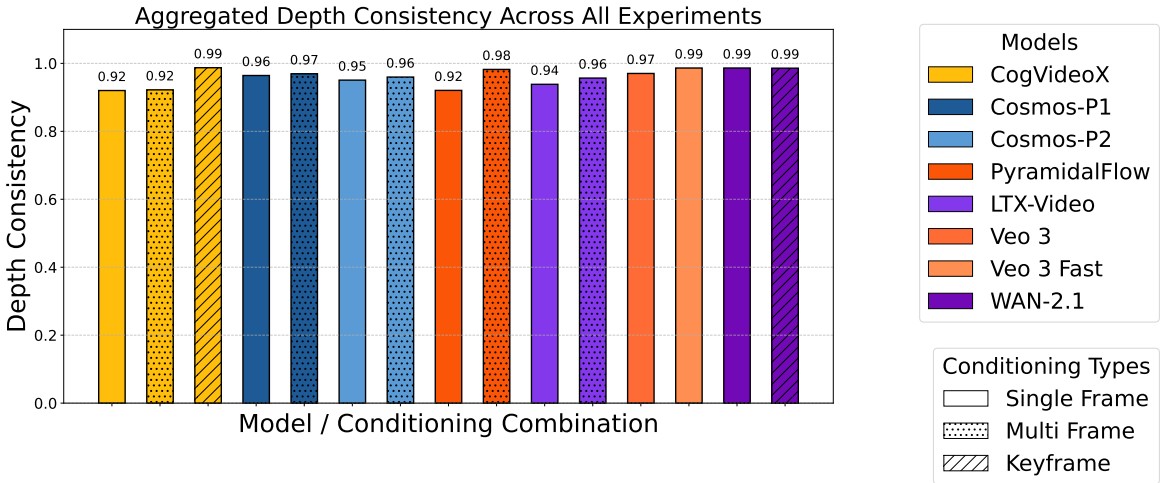

*Figure 16.* Average depth consistency for different video generation models across all studied experiments.

where:

- $y_i$ is the true value at timestep $i$,

- $\hat{y}_i$ is the predicted value at timestep $i$,

- $\bar{y}$ is the mean of the ground truth values, defined as:

$$\bar{y} = \frac{1}{N} \sum_{i=1}^{N} y_i, \tag{16}$$

- $\frac{1}{N} \sum_{i=1}^{N} (y_i - \hat{y}_i)^2$ represents the MSE between the predicted and ground truth trajectories,

- $\frac{1}{N} \sum_{i=1}^{N} (y_i - \bar{y})^2 = \sigma^2$ represents the variance of the ground truth trajectory.

To ensure robustness, the predicted trajectory is compared against the interpolated ground truth values. Depending on the experiment, we address physical consistency by quantifying how well the learned solution adheres to the underlying physical equation. This is quantified using the physics loss, which penalizes deviations from the expected dynamics. We set physics loss weight $\lambda = 1$ for simple systems (freefall, projectile, spring) and $\lambda = 5$ for chaotic systems (double pendulum) to prevent overfitting to noise. These values are retrieved based with validating on the real-world split; and never tuned on the generated videos. This ensures that changes in Dynamical Scores reflect genuine differences in the physical plausibility of the generated content, not artifacts of the evaluation procedure. During training, each PINN is optimized using the Adam optimizer with a learning rate of $10^{-3}$ for 200,000 iterations. The network used for all experiments, consists of two hidden layers of 20 neurons, with $\tanh$ as activation functions. The final score is defined as $S_{dyn} = \max(1 - \text{NMSE}, 0)$. Similarly to the Physical Invariance score, in cases when the original trajectory is discarded, the score is assigned to a minimal value equal to zero.

**PINN architecture ablation** To verify that the Dynamical score is insensitive to the specific PINN architecture, we sweep the number of hidden layers (1 to 4) and the hidden dimension (8 to 128) for the falling ball, pendulum, and projectile experiments. Table 3 reports the score we use in the paper alongside the best- and worst-performing configurations in the sweep. For simple dynamics (falling ball) the score stays nearly flat even for the smallest network (1 hidden layer, dimension 8). For more complex dynamics (pendulum, projectile) the score grows with capacity but plateaus early, and the gap between the best configuration and our paper architecture is negligible – confirming that our chosen architecture (two hidden layers of 20 neurons) has sufficient capacity.

| Experiment | Paper score | Best score | Worst score |
|---|---|---|---|
| Falling ball | 0.975 | 0.99 | 0.98 |
| Pendulum | 0.953 | 0.95 | 0.16 |
| Projectile | 0.875 | 0.86 | 0.09 |

*Table 3.* PINN architecture ablation over hidden layers (1–4) and hidden dimension (8–128). The paper configuration is within a negligible margin of the best-performing one.

**Manual Specification vs. Symbolic Discovery.** While integrating data-driven symbolic discovery could theoretically scale Morpheus to unknown physical systems, we deliberately avoid and prioritize reliability in the version of our paper. Symbolic discovery algorithms can be sensitive to noise and require large data, introducing confounding factors. For instance a low score could be attributed from a failure in discovery. By manually specifying the governing ODEs for known experimental setups, we treat the equation of motion as an immutable ground truth. In a future version one could bridge this gap with learning a *library of laws*approach, where the physical prior is selected with some heuristic or a simple classifier network from a predefined set.

### D.6. Comparison with VLM-based evaluators

We benchmark our collected data against VideoPHY2-AUTOEVAL (Bansal et al., 2026), a state-of-the-art VLM fine-tuned for physical commonsense, covering both our real-world recordings and the generations from nine models. All generated videos scored here were conditioned on the original real-world frames rather than the COSMOS-Transfer style-augmented initializations (App. B); we omit the augmented split here for brevity, but observed similar patterns. We evaluate two standard axes: *Physical Commonsense* (PC), a 1–5 rating of intuitive plausibility, and *Physical Rules* (PR), an indicator of whether a candidate physical rule is followed (1), violated (0), or indeterminate (2). Following the VideoPHY-2 methodology, we also report the *Joint Physical Consistency* (JPC) pass rate, requiring $PC \geq 4$ and $PR = 1$. Aggregate real-vs-generated results are in Table 4; per-model and per-category breakdowns are in Table 5 and Table 6. All reported 95% Confidence Intervals (CI) are computed as estimate $\pm 1.96$ SE: for the mean PC score SE $= \sigma/\sqrt{N}$ (sample standard deviation $\sigma$); for the PR-adherence and JPC pass rates we use Wald intervals with SE $= \sqrt{\hat{p}(1-\hat{p})/N}$ for a pass proportion $\hat{p}$. We treat two estimates as significantly different when their 95% CIs do not overlap.

| Source | PC (1–5) | PR adher. (PR = 1) | JPC pass |
|---|---|---|---|
| Real-World | 3.71 [3.57, 3.85] | 58.5% [49.8, 67.2] | 23.6% [16.1, 31.1] |
| Generated | 3.53 [3.49, 3.56] | 51.1% [48.5, 53.8] | 13.9% [12.1, 15.8] |

*Table 4.* VideoPHY2-AUTOEVAL (Bansal et al., 2026) on our collected data (95% CIs; **PC**: Physical Commonsense score (1–5); **PR**: Physical Rules adherence rate (fraction of videos with PR = 1); **JPC**: Joint Physical Consistency pass rate (fraction with PC $\geq 4$ and PR = 1). The real-world row averages over all experiments in the real-world split; the generated row averages over all models and experiments. The VLM fails to assign ceiling scores to real physics and barely separates real from generated, with no significant gap in rule adherence.

The VLM only weakly orders models by physical quality and frequently mis-ranks them: the highest mean PC score goes to Kling-Turbo (3.69) and PyramidalFlow (3.66), yet PyramidalFlow has the second-lowest rule adherence (43.8%). Most tellingly, the per-category analysis (Table 6) shows the evaluator assigning 0% JPC to clean real-world projectiles, bouncing balls, collisions, sliding books, and rolling cans, while assigning generated videos of the same phenomena comparable or higher pass rates. Across the full dataset, 42.3% of videos pass the holistic PC$\geq 4$ criterion but only 14.7% also satisfy the explicit rule check – i.e., VGMs routinely produce videos that look plausible to a VLM while breaking specific physical laws.

The per-phenomenon picture exposes the failure even more sharply. On the *Physical Commonsense* axis, the evaluator does not separate clean physics from generated physics: it assigns the complex double pendulum a *higher* mean PC to generations (3.46) than to the real-world ground truth (3.30), and rates elementary dynamics such as rolling cans and falling apples near-identically and only marginally (around 3.0) regardless of source. On the *Physical Rules* axis the hallucinated violations are starker still: the VLM assigns a 0% adherence rate to clean real-world bouncing balls and to varied-mass collisions (a light ball striking a heavier one), and penalizes real ground-truth physics below generated outputs for both the holonomic pendulum (real 77.8% vs. generated 100.0%) and sliding books (real 40.0% vs. generated 82.8%). An evaluator

that cannot confidently certify that a real video obeys physical laws is fundamentally unsuited to benchmarking generative models, which is precisely what motivates **Morpheus**'s physics-informed scoring.

| Model | PC (1–5) | PR adher. (PR = 1) | JPC pass |
|---|---|---|---|
| Veo3-fast | 3.56 | 60.4% | 25.0% |
| Kling-Turbo | 3.69 | 56.2% | 22.9% |
| Veo3 | 3.65 | 62.5% | 18.8% |
| COSMOS-predict2 | 3.55 | 52.6% | 17.2% |
| COSMOS-predict1 | 3.52 | 46.4% | 14.1% |
| WAN-2.1 | 3.60 | 51.6% | 13.0% |
| CogVideo | 3.35 | 54.9% | 12.2% |
| PyramidalFlow | 3.66 | 43.8% | 12.0% |
| LTX | 3.48 | 49.5% | 9.9% |

*Table 5.* VideoPHY2-AUTOEVAL (Bansal et al., 2026) scores per generated model (averaged across all experiment categories), sorted by JPC pass rate. .The VLM's ordering does not track physical quality (e.g., PyramidalFlow ranks high on PC but low on rule adherence).

| Experiment | Real-world JPC | Generated JPC |
|---|---|---|
| falling ball | 100.0% | 34.5% |
| rolling orange | 100.0% | 39.1% |
| spring | 85.7% | 72.4% |
| double pendulum | 30.0% | 41.4% |
| holonomic pendulum | 27.8% | 13.8% |
| bouncing ball | 0.0% | 5.7% |
| collision (big hits small) | 0.0% | 4.6% |
| collision (equal) | 0.0% | 0.0% |
| collision (small hits big) | 0.0% | 3.4% |
| falling apple | 0.0% | 0.0% |
| falling marker | 0.0% | 0.0% |
| falling tape | 0.0% | 0.0% |
| projectile | 0.0% | 2.3% |
| rolling empty can | 0.0% | 0.0% |
| rolling full can | 0.0% | 0.0% |
| sliding book | 0.0% | 5.7% |

*Table 6.* Per-category JPC pass rate, real-world vs. generated. The VLM hallucinates violations in clean real-world physics, assigning $0\%$ to many widespread phenomena while rewarding generations of the same category.

### D.7. Trajectory matching with Physics-IQ

To quantify the limitation of trajectory-matching evaluation, we conduct a *real-vs-real* comparison: for each experiment we cross-compare all pairs of distinct real-world recordings of the same phenomenon using the Physics-IQ (Motamed et al., 2026) metrics (Spatiotemporal IoU, Spatial IoU, and Weighted Spatial IoU), and average over pairings per category. The category-level results are reported in Table 7. Here, the original falling experiments (apple, ball, marker, tape) are grouped into "Falling", the collision experiments (equal, small-hits-big, big-hits-small) into "Collision", and the rolling experiments (empty can, full can, orange) into "Rolling". Because both videos in every pair are clean physical realizations, a metric that measured physical correctness should score them near-perfectly; instead the overlaps are low across all categories, confirming that small unobservable differences in initial conditions, mass distribution, or friction make trajectory matching reject physically valid futures. By scoring functional adherence to governing ODEs and conserved invariants instead, **Morpheus** scores near-perfectly on the same real-world references.

### D.8. Robustness to trajectory-extraction noise

We quantify the sensitivity of the **Morpheus** scores to common tracking errors – temporal gaps and mask-boundary inconsistency – via two perturbations applied to extracted trajectories: (i) *Jitter*, adding zero-mean Gaussian noise with scale $\sigma \in \{0.5, 1.0\}$, and (ii) *Gaps*, randomly dropping up to $20\%$ of trajectory points. We perturb real-world and generated

| Experiment | Spatio-temporal IoU | Spatial IoU | Weighted Spatial IoU |
|---|---|---|---|
| Bouncing Ball | 0.100 | 0.274 | 0.169 |
| Collision | 0.021 | 0.184 | 0.112 |
| Double Pendulum | 0.087 | 0.561 | 0.408 |
| Falling | 0.164 | 0.396 | 0.342 |
| Holonomic Pendulum | 0.072 | 0.532 | 0.397 |
| Projectile | 0.010 | 0.040 | 0.034 |
| Rolling | 0.306 | 0.773 | 0.725 |
| Sliding Book | 0.189 | 0.240 | 0.202 |
| Spring | 0.189 | 0.298 | 0.233 |

*Table 7.* Trajectory-matching metrics (Motamed et al., 2026) when comparing two *real-world* recordings of the same phenomenon. Even perfect-vs-perfect physics yields low overlap, showing that trajectory matching penalizes valid but divergent realizations.

(Veo3, Kling) trajectories from the projectile experiment and re-score. Across all settings the total score deviates by $\leq 3\%$ in relative terms. Our smoothing block (App. C) reduces this further – e.g., from $2.6\%$ to $1.8\%$ for real-world videos under Jitter with $\sigma = 1.0$ – and for the Gaps perturbation the deviation does not exceed $1\%$. We attribute this robustness to the combination of the sliding window, trajectory smoothing, and PINN training, each of which mitigates noise and gaps.

### D.9. Sliding-window ablation

The Physical Invariance score is computed on the best-scoring $25\%$ sliding window of each trajectory (App. D.12). To test the sensitivity of model rankings to this choice, we re-score with the full ($100\%$) trajectory and report the resulting per-conditioning rankings in Table 8. As expected, full-trajectory scoring lowers absolute scores across all models and conditioning types (typically by 0.05–0.10), because it forces the inclusion of high-variance segments such as the exact moments of impact in bouncing or collisions, where discrete tracking noise and unmodeled real-world dissipation depress the invariance score. Crucially, the *relative* rankings are highly stable: the single-frame hierarchy is near-identical to Figure 7 (only a swap of the virtually tied Veo3-fast/COSMOS-predict2), COSMOS-predict2 retains its multi-frame lead, and the keyframe ranking is unchanged with WAN-2.1 leading. This confirms that the performance gaps are driven by physical capability, not by the scoring-window length.

| Rank | Single Frame | Multi Frame | Keyframe |
|---|---|---|---|
| 1 | Kling-2.5-Turbo (0.50) | COSMOS-predict2 (0.51) | WAN-2.1 (0.59) |
| 2 | Veo3-fast (0.42) | PyramidalFlow (0.41) | CogVideo (0.39) |
| 3 | COSMOS-predict2 (0.41) | COSMOS-predict1 (0.40) | – |
| 4 | Veo3 (0.39) | CogVideo (0.39) | – |
| 5 | WAN-2.1 (0.38) | LTX (0.32) | – |
| 6 | COSMOS-predict1 (0.31) | – | – |
| 7 | PyramidalFlow (0.30) | – | – |
| 8 | CogVideo (0.26) | – | – |
| 9 | LTX (0.23) | – | – |

*Table 8.* Model rankings under full-trajectory ($100\%$ window) scoring. Absolute scores drop relative to the $25\%$ window, but the relative hierarchy is preserved across all conditioning types.

### D.10. Human alignment and filter validation

**Alignment with human judgment**  To verify that **Morpheus** scores track human perception of physical plausibility, three judges independently rated 65 generated videos on a 1–5 scale, marking outright generation failures for discard; we then averaged the per-video ratings. Since physical scores are only meaningful for videos that are not rejected as outright generation failures, we measure alignment on the 42 videos retained as valid by *both* the automatic discard filter and the human majority vote (the true-negative set of Fig. 19); these span multiple video generators (e.g., COSMOS-predict2, Kling, Veo3) under single- and multi-frame conditioning across our Newtonian experiments. On this set, the total **Morpheus** score attains a Spearman rank correlation of $\rho = 0.708$ ($p < 10^{-3}$) with the averaged human ratings (Fig. 17), indicating strong agreement.

Decomposing the total score into its two components (Fig. 18) reveals the following. The *dynamical* score, which captures motion plausibility, is strongly correlated with human judgment ($\rho = 0.768$, $p < 10^{-3}$), whereas the *physical-invariance* score, which measures the conservation of energy, momentum, and period, shows no significant correlation ($\rho = -0.157$, $p = 0.32$). This is expected and in fact reinforces the motivation for **Morpheus**: human raters (like the VLM evaluator of App. D.6) readily perceive whether motion *looks* right, but cannot reliably detect violations of physical invariants, which are subtle and quantitative. The invariance score is therefore validated against physics – real-world videos score $\geq 0.90$ – rather than against human perception. Human ratings are thus complementary to, not a replacement for, our physics-informed scores: a physics-based metric can in principle reward a dynamically correct but visually unusual video, whereas our goal is precise physical-consistency evaluation rather than visual plausibility (the latter being the focus of benchmarks such as VBench (Huang et al., 2024)).

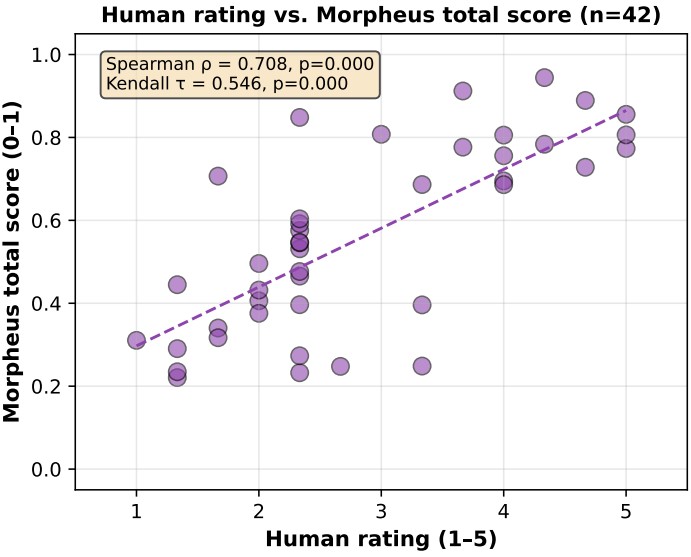

*Figure 17.* Human ratings vs. the total **Morpheus** score on the 42 videos retained as valid by both the automatic discard filter and the human majority vote. The total score correlates strongly with human judgment (Spearman $\rho = 0.708$, $p < 10^{-3}$).

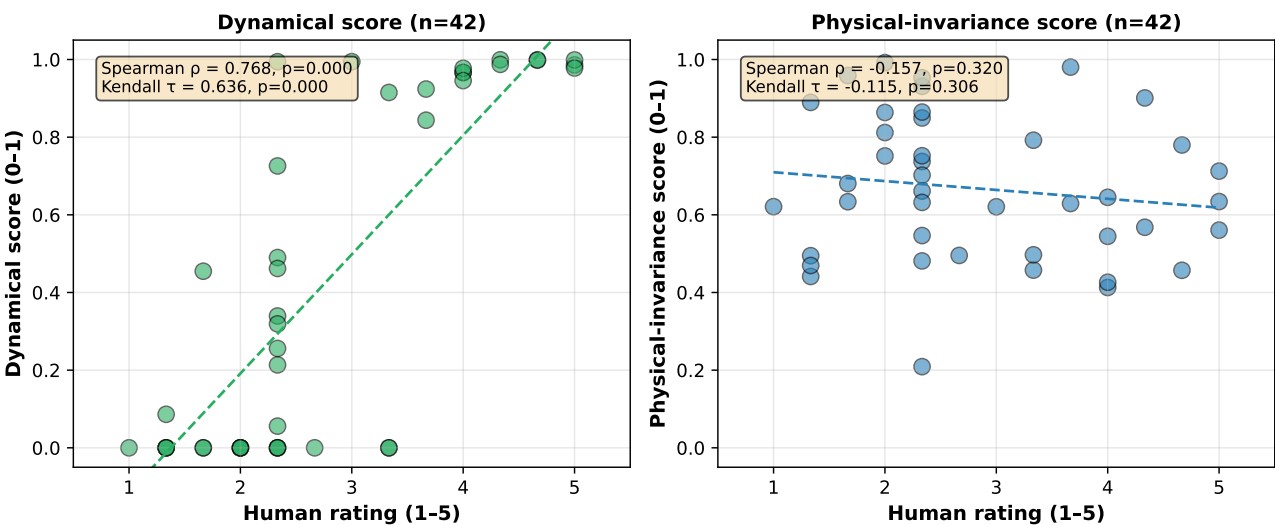

*Figure 18.* Sub-score decomposition of the human-alignment study ($n = 42$). **Left:** the dynamical score tracks human ratings closely ($\rho = 0.768$). **Right:** the physical-invariance score is uncorrelated with human ratings ($\rho = -0.157$, not significant), confirming that human raters perceive motion plausibility but not violations of physical invariants – precisely the gap that an automatic physics-based metric is needed to fill.

**Filtering validation** A potential concern is that the discard filter (App. D.1) is overly strict and removes valid generations, inflating discard rates and underestimating the models performance. To test this, three human judges scored 65 generated videos each, with the final decision by majority vote, and we compared against the automatic filter (Fig. 19). The filter exhibits low false positives (only 3 human-pass videos discarded), so valid videos are rarely removed; the residual errors are predominantly false negatives (11 invalid videos passing through), meaning the filter is *conservative* and the reported discard rates may, if anything, underestimate the true rate of unusable generations. Consistently, real-world videos yield essentially zero discard rate. The filter attains Accuracy 0.785, Precision 0.750, and Recall 0.450.

**Discard Decision Confusion Matrix**

Accuracy: 0.785   Precision: 0.750   Recall: 0.450   F1: 0.563

*Figure 19.* Confusion matrix of the automatic discard filter against majority-vote human judgment on the 65 rated videos (Accuracy 0.785, Precision 0.750, Recall 0.450, F1 0.563). The low false-positive count (3) confirms the filter rarely removes valid generations; the residual errors are predominantly false negatives (11), i.e. the filter is conservative.

**Failure-mode breakdown** Inspecting the discarded samples, the dominant failure mode is *stillness* (objects fail to move), followed by *disappearance* (object-permanence failure) and *duplication* (spurious extra copies of the object). These are visual-generation and object-tracking failures that occur *before* physical scoring; physical reasoning is evaluated only on the videos that pass filtering. Together with the trajectory-extraction robustness analysis (App. D.8), this indicates the discard rates reflect genuine generation failures rather than artifacts of the tracking pipeline.

### D.11. Physical Invariances

**Falling Ball** For falling balls, energy must be conserved between consecutive bouncing points. Additionally, according to Newton's second law:

$$F = ma$$

In free fall, gravity is the sole force acting on the object, resulting in constant acceleration. Assuming that the gravitational field is uniform in space and time, we have $F = mg$, which means that $a = g$, so the acceleration should stay constant. Therefore, in this part, we introduce three quantitative metrics to assess trajectory physics: the Energy Conservation score (ES), which measures energy conservation within a specified time window, and the Acceleration Conservation score (AS), which evaluates the consistency of acceleration during this interval, and the Horizontal Momentum Conservation score (MS), which measures the conservation of momentum.

*Table 9.* Conserved quantities for each physical experiment in an ideal case.

| Experiment Name | Assumption | Conserved Quantities |
|---|---|---|
| falling ball | no air resistance | energy, acceleration (gravity), horiz. momentum |
| projectile | no air resistance | energy, acceleration (gravity), horiz. momentum |
| bouncing ball | no air resistance | energy, acceleration (gravity), horiz. momentum |
| holonomic pendulum | low resistance | energy, period, pendulum length |
| sliding | uniform surface forces | acceleration |
| rolling | uniform surface forces | acceleration |
| elastic collision | perfect elasticity | total energy, linear momentum |
| spring-mass system | ideal Hookean spring | period |
| double pendulum | low resistance | total energy, two pendulums length |

The Energy Conservation score is calculated as follows. Given the mass of the ball to be $m$, the $g$ a freefall acceleration constant, kinetic energy:

$$T = \frac{1}{2}m|\vec{v}|^2 = \frac{1}{2}m(v_x^2 + v_y^2)$$

and potential energy:

$$V = mgh$$

where $h = y$. Total energy is the sum of two:

$$E = T + V = \frac{1}{2}m(v_x^2 + v_y^2) + mgy$$

From this formula, assuming the mass of the ball is constant in time, we get:

$$\frac{E}{m} = \frac{1}{2}(v_x^2 + v_y^2) + gy = \text{const} \tag{17}$$

The calculation of the Acceleration Conservation score is self-evident:

$$a = \text{const} \tag{18}$$

The conservation of horizontal momentum arises from the fact that the only force acting on the ball is gravity, which is pointed downwards:

$$p_x = mV_x = \text{const}$$

and analogous to the energy, we deduce:

$$\frac{p_x}{m} = V_x = \text{const} \tag{19}$$

We provide some examples of estimated invariants in Fig. 30.

**Projectile** For projectile motion, we analyze the same physical invariants as in the falling ball experiment. Throughout the projectile's trajectory, neglecting the air resistance, energy, acceleration, and horizontal momentum should be conserved. The calculations for energy and acceleration follow the same methodology as in the falling ball analysis.

**Holonomic Pendulum** For the holonomic pendulum, let's first examine energy conservation. Energy in the ideal (frictionless) case:

$$H = T + V = \frac{p_\theta^2}{2mL^2} + mgL(1 - \cos\theta)$$

where $\theta$ is the angular displacement, $l$ is the length of the pendulum, $g$ is the gravitational acceleration, and $p_\theta = mL^2\dot{\theta}$ is the momentum.

In this case, the equation we obtain is:

$$\ddot{\theta} + \frac{g}{l}\sin\theta = 0$$

Since our real-world pendulum experiments were conducted in a laboratory environment, friction causes energy attenuation over time. We quantify this energy loss by measuring both its range and rate of decline, establishing these as upper bounds for evaluating generated videos. To be specific, the holonomic pendulum with friction can be expressed as

$$\ddot{\theta} + \frac{b}{m}\dot{\theta} + \frac{g}{l}\sin\theta = 0 \tag{20}$$

where $b$ is the damping coefficient, $m$ is the bob mass, and $\frac{b}{m}\dot{\theta}$ represents the damping force term. The energy decay over time:

$$\frac{dE}{dt} = -b(\dot{\theta})^2 \tag{21}$$

In our experiments, we assume that the energy loss can be ignored for a short time period, meaning that we can apply the Energy Conservation score.

The period of holonomic pendulum with friction with a small amplitude can be expressed as

$$T = 2\pi\sqrt{\frac{l}{g}}\sqrt{1 - \left(\frac{b}{2m\omega_0}\right)^2} \tag{22}$$

where $\omega_0 = \sqrt{\frac{g}{l}}$ is the natural angular frequency without damping. When the damping is small ($b \ll m\omega_0$), the period approaches that of an undamped pendulum $T_0 = 2\pi\sqrt{\frac{l}{g}}$. We observe this regime in our experiments and propose to use the Period Conservation score (PC).

For the holonomic pendulum, it is obvious that the length of the pendulum $l$ remains constant throughout the experiment, as a holonomic constraint of the system. Therefore, we also consider the pendulum length as a physical invariant.

### D.12. Physical Score Scaling

When we obtain the physical invariant value $C$, we calculate the relative standard deviation over time:

$$C_{\bar{\sigma}} = C_\sigma / C_{mean} \tag{23}$$

To ensure that the score is within the range $[0, 1]$, we design the Physical score, derived from the invariant, as follows

$$S = \frac{1}{1 + \alpha * C_{\bar{\sigma}}} \tag{24}$$

Where $\alpha$ is a normalization factor. In the experiment, we set it to 1.0.

Two critical considerations emerge during the score calculation process. First, The time window must be carefully selected. For each trajectory, we partition it using a sliding time window and select the highest score among all segments as the overall score of the trajectory. This approach addresses a key challenge in real-world experiments such as bouncing balls, where fluctuations near bouncing points create large standard deviations and low scores. Using the highest score across all segments, we effectively capture the most stable portion of the trajectory.

In our experiments, we set the time window length equal to 25% of the total trajectory duration. Specifically, we use $t_{window} = L_{trajectory}/4$. In addition, proper scaling is essential: since the trajectory coordinates are recorded in pixel space rather than real-world 3D coordinates, a precise coordinate transformation to physical units is required. In particular, improper scaling can significantly impact the total energy calculations. Third, we need to be careful not to choose the range when the mean of the selected physical invariant is near zero. The absolute value of the mean of the selected physical invariant should be equal to or greater than a threshold of 10 times the standard deviation.

$$C_{threshold} = 10 * C_\sigma \tag{25}$$

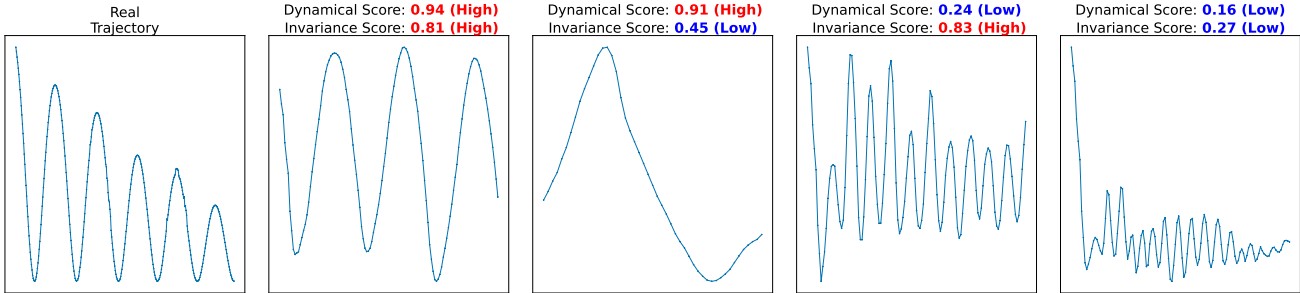

*Figure 20.* Real pendulum trajectory alongside four cases of generated videos, demonstrating how different combinations of dynamical and physical invariance scores appear in practice.

so it can be neglected that the influence of mean energy/acceleration is near zero. In the experiment, if $|C_{mean}| \geq C_{threshold}$, we calculate the Physical score as defined in Eq. 24. Otherwise, we use the following Eq. 26 which takes the absolute standard deviation rather than the relative standard deviation.

$$S = \frac{1}{1 + \alpha * C_\sigma} \tag{26}$$

This method has two key limitations. First, scores are highly sensitive to the choice of time window size. Larger time windows tend to yield lower scores as they encompass more fluctuations in the trajectory. Thus, we kept the time window constant between the evaluation of different generations of models. Second, the method may fail to detect unphysical behavior in generated videos where objects remain stationary for long periods; however, this is addressed by discarding videos due to stillness.

## E. Extended Limitations

The main focus of our benchmark is towards a set of Newtonian scenarios, which take place under a controlled environment and static camera conditions. While this enables clean invariants and can be reproducible, it limits a full scale physical coverage. Additionally, ambiguity is introduced (e.g. distortion of lens, unknown scale, etc.) because all of our estimates come from 2D pixel trajectories without any camera calibration. At the same time, we rely on assumptions such as negligible air resistance and friction. Unmodeled forces, such as air drag, friction and rotational kinetic energy could cause violations, which do not reflect generations that are wrong but rather evaluator mismatch. We perform trajectory extraction from generated videos using fixed first/last frame or short multi-frame conditioning that cannot specify initial conditions fully, such as the mass of an object or the coefficient of the spring. Moreover, a physically plausible generation can be heavily penalized when some of the assumptions are violated, leading to misleading results. Similarly, one single invariant can look really "good", but the generated video could violate multiple laws, making it physically implausible.

Errors in segmentation and tracking introduce noise. Additionally, we are only considering short clips, and the model might drift away from physical laws over a longer temporal horizon, which our current metrics would not be capable of capturing. Finally, it should be noted that while our scores remain proxies rather than direct measurements of underlying physical plausibility, that would require falsifiable and controllable VGM generation. Prior benchmarks that rely on judgments from humans or VLMs are prone to hallucinations and often miss slight inconsistencies. In contrast, our scores avoid these issues but still need to be interpreted jointly to give a much more reliable picture. With that in mind, Morpheus should be treated as a focused and reproducible stress test for core physical laws and not a benchmark that can test and evaluate all physical dynamics that might take place in a single video.

## F. Video Generative Models Details

At their core, latent video generative models often utilize a combination of a 3D Variational Autoencoder (VAE) (Kingma & Welling, 2014; Lin et al., 2024) to tokenize individual frames, a text encoder, such as T5 (Raffel et al., 2020) to encode frames into latent. During training a noisy latent is produced by the forward diffusion process. This latent is then processed by a parametrized model, either a transformer model (Yang et al., 2023) or a U-Net (Ronneberger et al., 2015; Zhou et al., 2022; Bar-Tal et al., 2024) resulting in a patchified sequence of visual tokens, in the case of the former type of model.

Depending on the model architecture, different input modalities can be handled, such as text-to-video, image-to-video, text+image-to-video, and sometimes video continuation regimes facilitating both open-domain and controlled generation scenarios (Yang et al., 2025; Kong et al., 2024; Agarwal et al., 2025; Brooks et al., 2024). Although some state-of-the-art video generation models adopt an autoregressive framework, predicting frames sequentially based on prior outputs (Weng et al., 2024; Deng et al., 2024; Weissenborn et al., 2020), many others utilize non-autoregressive approaches to generate frames simultaneously (Yang et al., 2025; Kong et al., 2024; Xing et al., 2024). In Table 10, we specify the parameters of particular models used in our benchmark, along with it's architectural design choices. To faithfully obtain the best generation outcome, we use the best hyperparameters reported for each model.

| Model [Params.] | Resolution | Number of Video Frames | Guidance Scale | Sampling Steps |
|---|---|---|---|---|
| CogVideoX [5B] | 960 x 768 | 84 | 6.0 | 50 |
| Cosmos-Predict 1 [14B] | 1280 × 704 | 121 | 7.0 | 35 |
| Cosmos-Predict 2 [14B] | 1280 × 704 | 93 | 7.0 | 35 |
| WAN2.1 [14B] | 1280 × 720 | 121 | 5.0 | 40 |
| LTX-Video [13B] | 960x736 | 81 | 3.0 | 50 |
| PyramidalFlow[12B] | 1280 x 768 | 121 | 4.0 | 10 |

*Table 10.* Details of video generation models adopted in our benchmark study, including their resolution, number of video frames, guidance scale, and sampling steps.

## G. Prompts for Video Generation Models

For each experiment, we carefully designed a prompt that describes the physical setup and motion of the experiment being conducted. For example, in the falling ball experiment, the prompt specifies that the ball falls and makes contact with the table below. Similarly, in the projectile experiment, we describe how the ball is launched at a slight upward angle and follows a natural parabolic trajectory. We enhance these prompts using an internally provided upsampler or ChatGLM if no upsampler is provided to incorporate more detailed scene descriptions and contextual elements derived from the reference images. All prompts are shown in Table 11 and Table 12.

| Experiment Name | Base Prompt | Enhanced Prompt |
|---|---|---|
| **Falling Ball** | Orange ping-pong ball falling down and making impact with the table surface below. Fixed camera view, no camera movement. | A ping-pong ball is captured in mid-air, suspended above a laboratory table, poised to make contact with the surface below. The ball's descent is governed by the force of gravity, creating an arc that suggests a controlled experiment in progress. The backdrop is a stark, clinical room with a neutral palette, punctuated by the sterile lines of a metal frame and the functional design of a nearby cabinet. The lighting is subdued, casting a soft glow that highlights the ball's trajectory and the anticipation of impact. The table beneath the ball is marked with faint lines, perhaps indicating measurements or guidelines for the experiment. As the ball continues its downward journey, it will likely bounce off the table, adding a dynamic element to the scene and marking the conclusion of this controlled descent. Fixed camera view, no camera movement. |
| **Projectile** | A single, small 3D-printed ball, dark gray in color, is launched from a plastic, small-scale ramp with a slight upward angle. The ball follows a natural, smooth, arcing trajectory upward and then downward, continuing along that arc until it exits the right side of the video frame. The video should accurately simulate the ball's motion under standard earth gravity, showing a clear parabolic arc. The video should emphasize a smooth and realistic physics-based movement of the ball without any sudden changes in speed. The ball should not bounce or collide with any objects in the scene. Fixed camera view, no camera movement. | In a meticulously crafted scene, a solitary, dark gray 3D-printed ball, with its sleek, spherical form, is propelled from a plastic ramp that slopes gently upward. The ball, weighing a mere fraction of a kilogram, is captured in high-definition, showcasing every nuance of its motion. As it leaves the ramp's edge, the ball arcs gracefully into the air, its trajectory a perfect parabola that mirrors the laws of physics under standard earth gravity. The video's frame follows the ball's smooth ascent and descent, highlighting the ball's consistent speed and the absence of any sudden accelerations or decelerations. The scene remains unobstructed, ensuring that the ball's journey is uninterrupted by any external forces, save for the pull of gravity, resulting in a visually stunning and scientifically accurate demonstration of a parabolic motion. Fixed camera view, no camera movement. |
| **Rolling Can** | An empty soda can rolls down a wooden surface. Experiment carried out in a laboratory controlled environment. Fixed camera view, no camera movement. | An empty aluminum red soda can rolls steadily down an inclined wooden board in a controlled laboratory environment. This motion should be depicted with physical realism, accurately simulating the key dynamics: the can accelerates under gravity, with its combined rotational and translational movement appearing authentic and governed by the frictional interaction with the wooden surface, ensuring a physically plausible rolling movement. Fixed camera view, no camera movement. |
| **Sliding Book** | A book slides down a wooden surface. Experiment carried out in a laboratory controlled environment. Fixed camera view, no camera movement. | A book slides steadily down an inclined wooden board in a controlled laboratory environment. This motion should be depicted with physical realism, accurately simulating the key dynamics: the book accelerates due to the component of gravity acting along the incline, opposed by kinetic friction from the wooden surface. Its movement should be purely translational, maintaining consistent contact with the board and without any significant rotation or tumbling, ensuring a physically plausible descent. Fixed camera view, no camera movement. |
| **Holonomic Pendulum** | A single pendulum moving retrogressively back and forth. At the bottom of a pendulum, there is a ball attached to it. The pendulum is holonomic. Fixed camera view, no camera movement. | A pendulum with a spherical ball attached swings back and forth in a controlled manner, its motion captured in a moment of retrograde swing. The pendulum's arm, likely made of metal, extends horizontally from a stand, connected to a pivot point that allows for rotational movement. The ball, positioned at the lower end of the pendulum, appears to be in motion, indicating the pendulum's swing. The environment suggests a laboratory or testing setting, with a backdrop of technical apparatus and equipment, and the lighting is artificial, casting a uniform glow over the scene. The pendulum's movement, while currently in a retrogressive swing, could potentially change direction, continuing its oscillatory motion. Fixed camera view, no camera movement. |

*Table 11.* Base and enhanced textual prompts used for video generation experiments: Falling Ball, Projectile, Rolling Can, Sliding Book, Holonomic Pendulum. Enhanced prompts are generated using corresponding upsampler VLMs such as ChatGLM (Zeng et al., 2024) and incorporate more detailed scene descriptions and contextual elements derived from the reference images. Slight modifications to these prompts were made in case of different variants such as object type.

| Experiment Name | Base Prompt | Enhanced Prompt |
|---|---|---|
| **Double Pendulum** | Double pendulum, consisting of a purple and an orange segment. Each segment moves independently. Fixed camera view, no camera movement. | In a meticulously arranged laboratory setting, a double pendulum setup swings gracefully, each pendulum segment adhering to the immutable laws of physics. The upper pendulum, a sleek purple rod, contrasts strikingly with the lower orange rod, both suspended from a sturdy, metallic frame. The room is bathed in soft, ambient light, casting subtle shadows that accentuate the pendulums' arcs. The scene captures the intricate dance of the pendulums, their movements a mesmerizing testament to the natural order, with each swing a silent symphony of motion and balance. Fixed camera view, no camera movement. |
| **Bouncing Ball** | A single orange ping pong ball bounces vertically as a result of making impact with the table after being in free fall. The ball starts in the center of the frame, and moves upwards. Fixed camera view, no camera movement. | A solitary orange ping pong ball, with its vibrant hue standing out against the stark white of the table, plummets from the center of the frame. As the ball bounces upwards, it arcs gracefully, the trajectory a perfect parabola. The frame remains centered, emphasizing the ball's solitary dance of motion and the physics of its rebound. Fixed camera view, no camera movement. |
| **Collision** | Generate a realistic video of two metallic spheres colliding. In the first frames the leftmost sphere in the video is moving towards the static one. At the moment of contact the ball in motion transfer its kinetic energy to the ball at rest. The two spheres have identical physical properties, namely material, shapes, masses. The momentum should be conserved. Fixed camera view, no camera movement. | In a high-definition, slow-motion sequence, two identical metallic spheres, each polished to a mirror-like finish, are meticulously positioned on a frictionless surface. The left sphere, propelled by an unseen force, hurtles towards the stationary sphere, its trajectory a perfect parabola. As the oncoming sphere approaches, the static sphere begins to subtly vibrate, a prelude to the impending collision. The moment of contact is captured in stunning detail, revealing the transfer of kinetic energy as the moving sphere's momentum is transferred to the stationary one. The spheres, crafted from the same dense material, maintain their spherical shapes and masses, ensuring the conservation of momentum throughout the impact. The scene is illuminated by a single, soft light source, casting long shadows and emphasizing the physics of the collision. Fixed camera view, no camera movement. |
| **Spring** | A single metallic slotted mass attached on a spring moving periodically up and down. Fixed camera view, no camera movement. | In a meticulously crafted scene, a solitary metallic slotted mass, polished to a gleaming silver, is ingeniously attached to a tensioned spring. The mass, weighing several pounds, oscillates with a fluid grace, its movement initiated by the subtle release of the spring. The camera captures the mass as it arcs upwards, the tension in the spring visible, before it gently descends with a rhythmic sway, the sound of its metallic slats clinking softly in the background. The scene is set against a stark white backdrop, emphasizing the stark contrast between the mass and the spring, and the smooth, periodic motion of the mechanical dance. Fixed camera view, no camera movement. |

*Table 12.* Base and enhanced textual prompts used for video generation experiments: Double Pendulum, Bouncing Ball, Collision, Spring. Enhanced prompts are generated using corresponding upsampler VLMs such as ChatGLM (Zeng et al., 2024) and incorporate more detailed scene descriptions and contextual elements derived from the reference images. Slight modifications to these prompts were made in case of different variants such as object type.

## H. Additional Analysis

In Figure 30, we present an additional visualization of the energy and acceleration conservation. In Figure 20, we visualize the difference between the object trajectories of real-world and generated videos. In Figure 30(a), we present the total, kinetic, and potential energy over time. As expected, the total energy dissipates with every new bounce while remaining nearly constant between bounces.

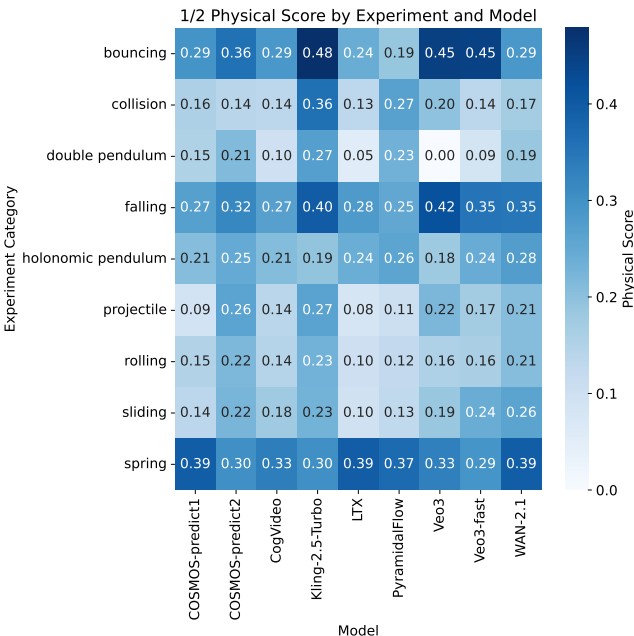

*Figure 21.* The distribution of the average physical invariance score across models and experiments.

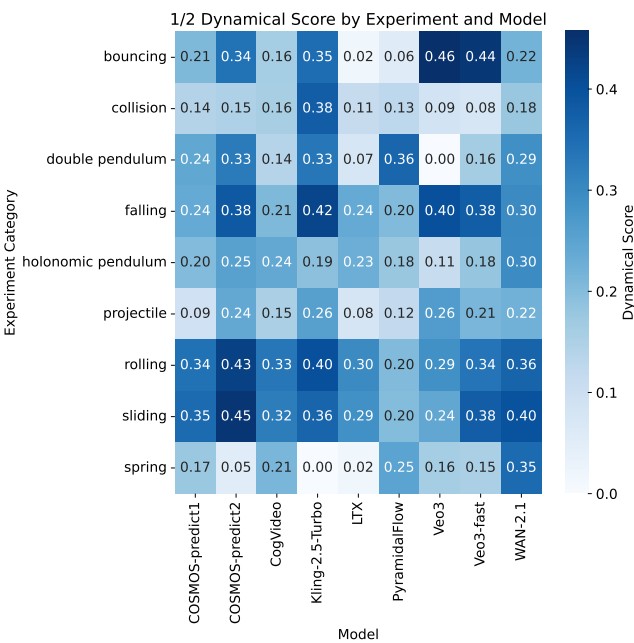

*Figure 22.* The distribution of the average dynamical score across models and experiments.

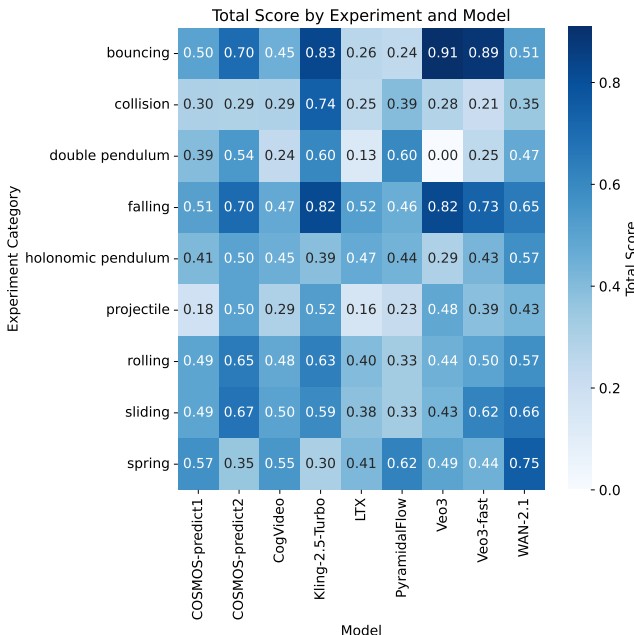

*Figure 23.* The distribution of the average total score across models and experiments.

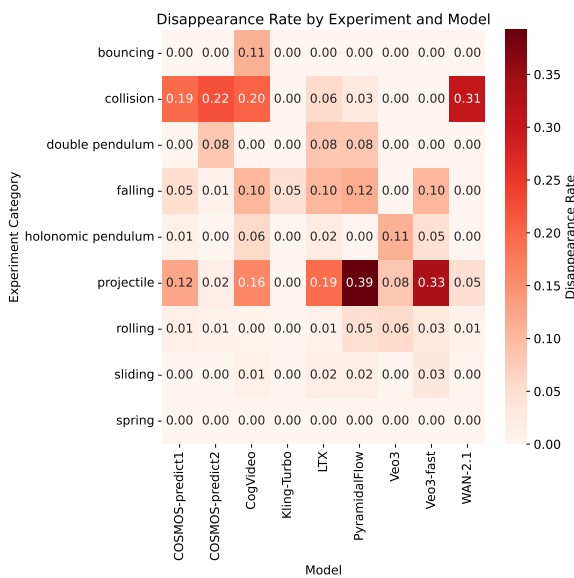

*Figure 24.* The distribution of the average discard rate due to the **objects' disappearance** across models and experiments.

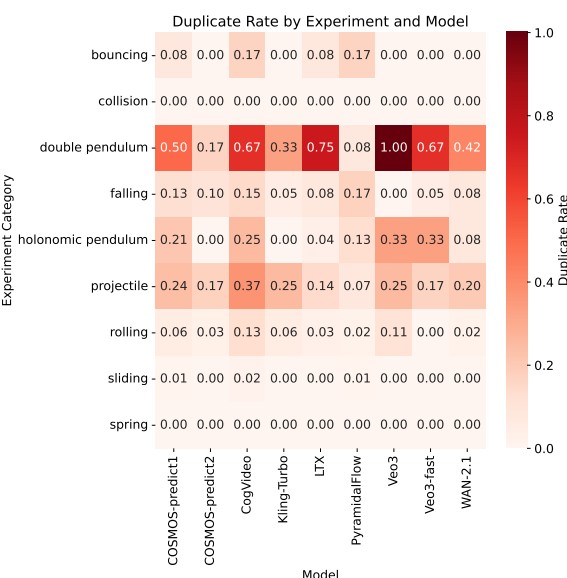

*Figure 25.* The distribution of the average discard rate due to the **objects' duplicates** across models and experiments.

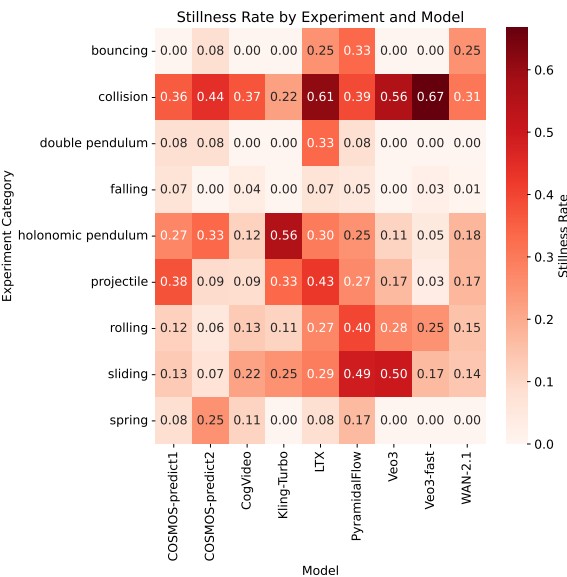

*Figure 26.* The distribution of the average discard rate due to **objects' stillness** across models and experiments.

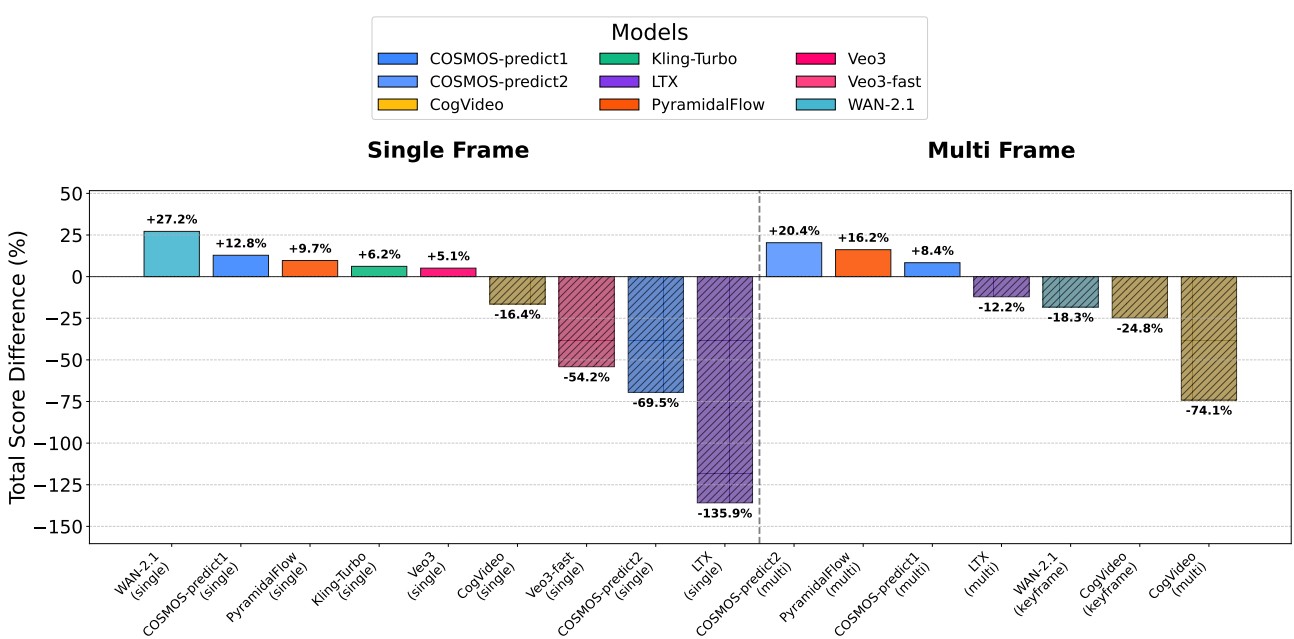

*Figure 27.* The relative change in scores evaluating on original data vs. on the augmented data.

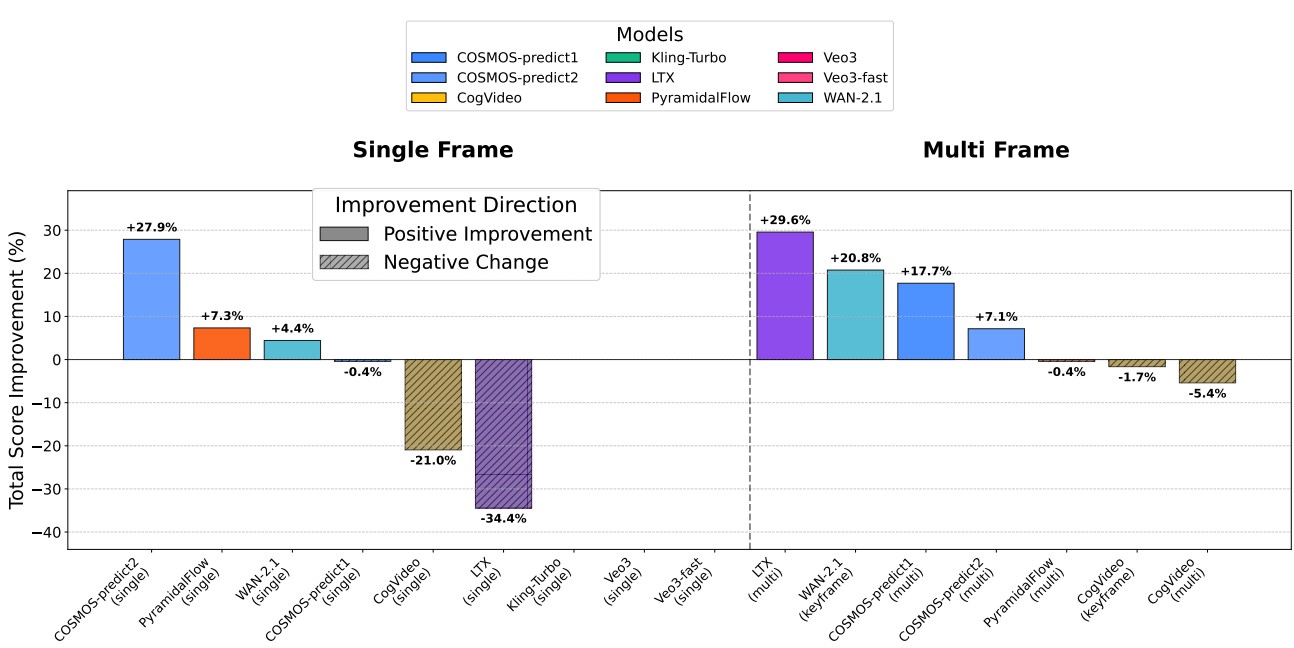

*Figure 28.* The relative change in scores for using an enhanced prompt vs the plain one.

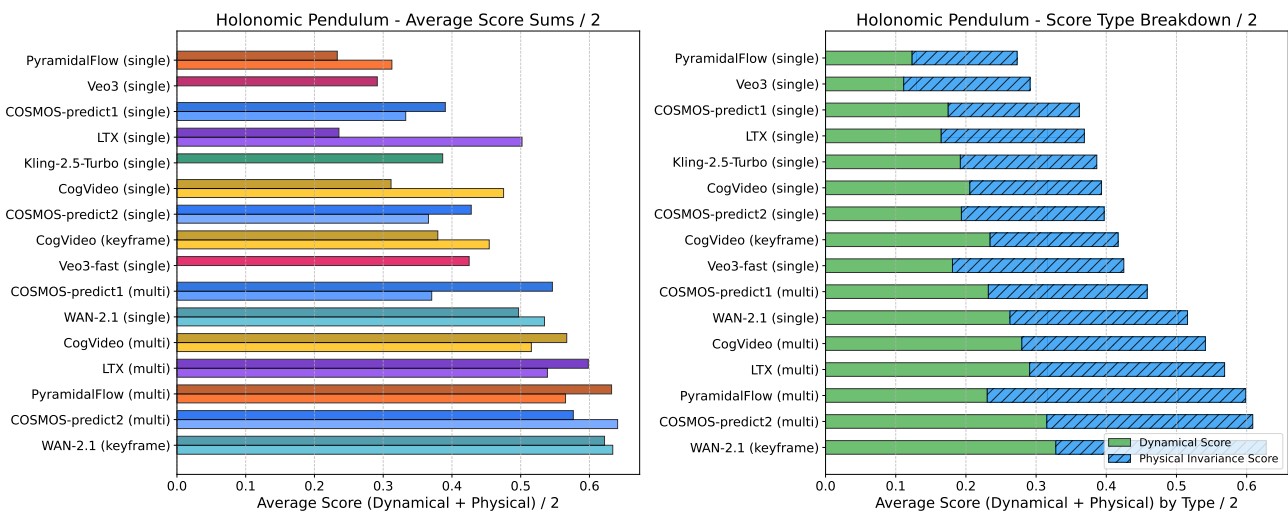

*Figure 29.* The scores for the holonomic pendulum experiment. On the left, darker colors denote *enhanced* textual prompt.

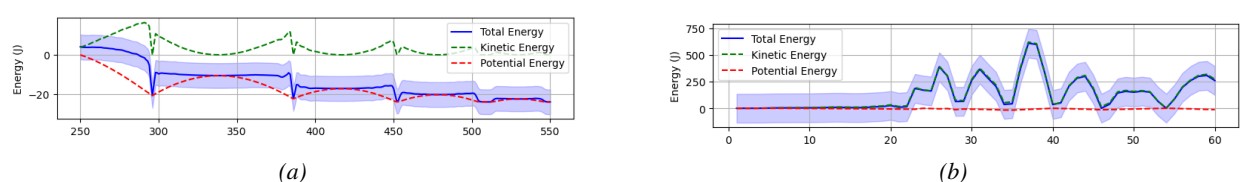

*(a)*                    *(b)*

*Figure 30.* Energy analysis of real-world and generated falling + bouncing ball videos: (a) Real-world video energy conservation (b) CogVideoX plain single frame generated video energy conservation

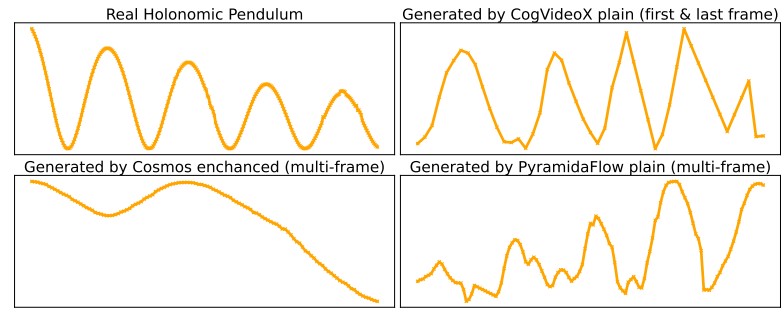

*Figure 31.* Real (top left) and generated trajectories for the holonomic pendulum.

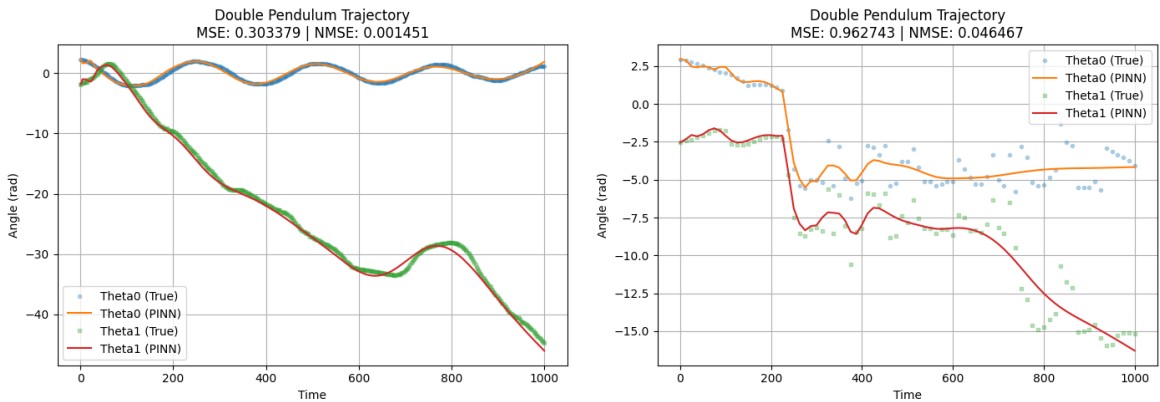

*Figure 32.* Real (left) and generated trajectory (right) for the double pendulum and corresponding fitting curve with PINN. While NMSE for generated trajectory is small 0.05, it is still 50 times worse that PINN with the same parameters fitted to real-world trajectory.

