# OpenReview forum: "Evaluating Newtonian Mechanics in Video Generative Models with Real Physical Systems"
_ICML.cc/2026/Conference — ICML 2026 regular_

### Official Review · Reviewer_Xw1q · 2026-03-11

**Soundness:** 2
**Presentation:** 3
**Significance:** 2
**Originality:** 2
**Overall Recommendation:** 4
**Confidence:** 2

**Summary:**

This paper introduces Morpheus, a benchmark and evaluation framework for testing whether video generative models respect Newtonian mechanics in controlled rigid-body scenes. The benchmark contains about 130 real-world videos across several physical setups, and evaluates generated videos using three metrics: discard rate, a PINN-based Dynamical score, and a Physical Invariance score based on conserved quantities. The authors evaluate several open- and closed-source video generation models under text, image, and video conditioning, and report that current models remain far from the physical fidelity observed in the real-world reference videos. The paper argues that physics-informed evaluation is more appropriate than human/VLM judgment or raw trajectory matching for this setting.

**Compliance With Llm Reviewing Policy:**

Affirmed.

**Final Justification:**

I appreciate the detailed rebuttal. Most of my concerns have been addressed, so I will maintain my original WA rating.

**Key Questions For Authors:**

- How sensitive are the model rankings to the invariant scoring protocol, especially the choice to take the best-scoring 25% window rather than scoring the full valid trajectory? If you used full-trajectory scoring, would the ranking in Figure 7 materially change?

**Limitations:**

yes

**Strengths And Weaknesses:**

**Strengths:**
1. The overall benchmark design is sensible for the stated goal. Restricting to controlled Newtonian rigid-body settings makes quantitative evaluation possible, which is a real advantage over broader but vaguer “physical plausibility” benchmarks.
2. The dataset comes from 9 controlled experiments (varied initial conditions) and metrics are validated with real videos. It evaluates several VGMs with systematic analyses of prompts and conditioning modes. Ablations isolate variable impacts to avoid cherry-picking.
3. The paper is generally readable, and the high-level visual intuition is communicated well.


**Weaknesses:**
1. The paper argues for objective physics-based evaluation via trajectory extraction and physics-informed scoring. However, it does not include a human study or correlation analysis to validate that the proposed scores track human perception of physical plausibility. Without this sanity check, it remains unclear whether the metric may penalize visually/plausibly physical generations (e.g., due to unmodeled effects or tracker noise) or over-reward outputs that fit the assumed dynamics while appearing visually implausible. This weakens confidence that the benchmark reflects “physical realism” as perceived by users and practitioners.
2. While the framework is well-motivated for conditioned generation settings (where the initial state is visually grounded and the target object(s) are explicitly identifiable), the paper does not present a concrete protocol for fair evaluation of pure text-to-video models. In T2V, the initial physical state (object identity, scale, pose, and implied velocities) can be ambiguous or under-specified, and it is unclear how the pipeline ensures consistent object selection, comparable initialization, and interpretable physics scoring across models. As a result, the benchmark’s applicability appears restricted to a subset of VGM modalities.
3. The core metric definition contains an apparent error. On Page 20, ($S_{dyn}=\min(1-\mathrm{NMSE},0)$) cannot produce the positive scores shown in Figure 7.

---

> ### Author Rebuttal · Authors · 2026-03-31
>
> We thank the reviewer for the thoughtful feedback and for pointing out these important issues.
>
> **On sensitivity to the sliding window size** We agree that this is an important question. To address it, we are adding an ablation over the sliding window size. We run the scoring with the 100% window (as suggested) and present the ranking below.
>
>
> | Rank  | Single Frame           | Multi Frame            | Keyframe        |
> | ----- | ---------------------- | ---------------------- | --------------- |
> | **1** | Kling-2.5-Turbo (0.50) | COSMOS-predict2 (0.51) | WAN-2.1 (0.59)  |
> | **2** | Veo3-fast (0.42)       | PyramidalFlow (0.41)   | CogVideo (0.39) |
> | **3** | COSMOS-predict2 (0.41) | COSMOS-predict1 (0.40) | (-)|
> | **4** | Veo3 (0.39)| CogVideo (0.39) | (-) |
> | **5** | WAN-2.1 (0.38) | LTX (0.32)| (-) |
> | **6** | COSMOS-predict1 (0.31) | (-)| (-)|
> | **7** | PyramidalFlow (0.30) | (-) | (-) |
> | **8** | CogVideo (0.26)| (-) | (-) |
> | **9** | LTX (0.23) | (-) | (-)|
>
> As expected, evaluating with full trajectory leads to a systemic drop in absolute scores across all models and conditioning types (typically by a margin of 0.05 to 0.10). This reduction validates our original rationale discussed in Appendix D.7: full-trajectory scoring forces the inclusion of periods with high variance, such as the exact moments of impact in bouncing or collisions. At these boundaries, discrete tracking noise and unmodeled real-world energy dissipation, naturally lower the physical invariance score.
>
> Crucially, despite the drop in absolute scores, the relative rankings of the models do not fluctuate significantly.
>
> **Single Frame:** The hierarchy remains almost identical to Figure 7. The only minor shift is a swap between Veo3-fast and COSMOS-predict2 for second and third best, though their scores remain virtually tied. The bottom six models maintain their exact original order.
>
> **Multi Frame:** COSMOS-predict2 maintains its definitive lead. A similarly minor swap occurs between PyramidalFlow and COSMOS-predict1 for second and third best, while the rest of the ranking holds.
>
> **Keyframe:** The rankings remain entirely unchanged, with WAN-2.1 maintaining a significant lead over CogVideo.
>
> The provided experiment confirms that the performance gaps between models are driven by their actual physical capabilities, not by hyperparameter sensitivity. This proves that while the 25% window is a useful calibration step to mitigate legitimate real-world noise, the Morpheus evaluation framework remains highly stable and objective regardless of the specific scoring window length.
>
> **On human score alignment.** We agree that the current version would benefit from an explicit sanity check against human judgments. To address this, we are adding correlation analysis with human ratings of physical plausibility. Each of 3 judges ranked 42 videos on the scale of 1 to 5, and the score was averaged afterwards. We observe the **Spearman rank correlation of 0.708** between the total score and the human judgement. We provide a visualization for it here: **[anonymous demo](https://anonymous.4open.science/w/dtoemq1iw0/)**
>
> We also note that real-world videos achieve near-perfect scores in our framework, which suggests that unmodeled real-world effects do not materially break the metric in practice, and we further validate the robustness of the tracking pipeline through perturbation-based analysis (see answer to Reviewer tzjf). We agree that a physics-based metric can in principle over-reward visually implausible but dynamically correct cases; however, our goal here is precise evaluation of physical consistency, which is complementary to existing benchmarks focused on visual plausibility (e.g. VBench [1]).
>
> **On extension to pure text-to-video.** We agree that pure text-to-video is more ambiguous and harder to evaluate fairly, since the initial state is not visually fixed and object identity/scale can be under-specified. We will clarify this limitation more explicitly. At the same time, we believe the framework can be extended to this setting using modern open-vocabulary detection/tracking: although the realized initial conditions may vary across generations, once the relevant object is identified and tracked, the resulting motion can still be evaluated against the same governing physical law. In this sense, the main additional challenge is robust object grounding/tracking, rather than the physics-based scoring itself. This also distinguishes our setting from trajectory-matching approaches, which require a fixed reference trajectory.
>
> **On the metric definition typo.** We thank the reviewer for catching this error. The expression on Page 24 contains a typo, and we will correct upon acceptance.
>
> [1]. Huang, Ziqi, et al. "Vbench: Comprehensive benchmark suite for video generative models." Proceedings of the IEEE/CVF Conference on Computer Vision and Pattern Recognition. 2024.

---

> > ### Author Rebuttal · Reviewer_Xw1q · 2026-04-04
> >
> > Thank you for the rebuttal. My concerns have been addressed. Good luck!

---

> > > ### Author Response · Authors · 2026-04-04
> > >
> > > Thank you for the positive feedback and for taking the time to review our rebuttal — we are glad the concerns have been addressed. If you feel the clarifications have strengthened the paper, we would be grateful if this could be reflected in the final score. If there are any remaining questions, we would be happy to address them.

---

### Official Review · Reviewer_mnZD · 2026-03-12

**Soundness:** 3
**Presentation:** 2
**Significance:** 2
**Originality:** 3
**Overall Recommendation:** 4
**Confidence:** 4

**Summary:**

This paper introduces Morpheus, a physics-informed evaluation framework for probing whether video generative models capture Newtonian rigid-body dynamics rather than merely producing visually plausible motion. Morpheus features 130 real-world videos capturing physical phenomena, guided by conservation laws.

**Compliance With Llm Reviewing Policy:**

Affirmed.

**Final Justification:**

Thanks for the detailed rebuttal, my concerns are largely addressed. I will keep the original WA rating.

**Key Questions For Authors:**

**Q1. Robustness of the Evaluation Pipeline**: Figure 6 shows high discard rates across all models, suggesting the trajectory extraction pipeline may be fragile. How sensitive the final scores are to different components of the pipeline?

**Q2. Failure Mode Analysis**: Discard rates may conflate multiple factors, such as visual generation failures, object persistence issues, tracking failures, and physical reasoning errors. Can the authors provide a clearer breakdown of these failure modes?

**Q3. Real-World Video Results**: Section 5.1 states that *real-world videos achieve Dynamical Scores of 0.96–0.99 and Physical Invariance Scores ≥0.90*, but these results do not appear in the tables. Could the authors clarify where these numbers come from or include them explicitly in the results?

**Q4. Metric Reporting**: Figure 7 reports only the average score.  It would be more informative if the individual metrics were reported separately.

**Limitations:**

Yes

**Strengths And Weaknesses:**

**Strengths**

1. The paper tackles an important question for the community, namely whether video generative models capture Newtonian rigid-body dynamics rather than merely producing visually plausible motion.
2. The use of real-world videos, instead of only synthetic simulations, is a meaningful strength. That makes the benchmark more relevant to embodied-AI use cases than many purely simulated alternatives.

**Weaknesses**

1. Figure 6 shows that the discard rates are high across all models, suggesting that the evaluation pipeline imposes very strict constraints on generated videos to obtain reliable trajectories. Such high discard rates likely reflect the fragility of the trajectory extraction pipeline, which relies on many prerequisite steps.
2. Beyond the cases shown in Figure 5, discard rates may conflate multiple factors, including visual generation failures, object persistence, tracking success, and physical reasoning. The authors should provide a clearer categorization of these failure modes.
3. Section 5.1 states that *“Real-world videos consistently serve as the upper bound for our evaluation, achieving high Dynamical Scores (0.96–0.99) and Physical Invariance Scores (≥0.90).”* However, these results do not appear in the tables. It is unclear whether this information was omitted.
4. Figure 7 reports only the average score. It would be more informative if the individual metrics were reported separately.

---

> ### Author Rebuttal · Authors · 2026-03-31
>
> We thank the reviewer for the thoughtful feedback and constructive suggestions.
>
> **Q1: Robustness of the trajectory extraction pipeline.** We agree that high discard rates would be concerning if they were primarily caused by an overly strict filtering pipeline. To test this, we conducted human evaluation of the filtering stage, with 3 human judges scoring 65 generated videos each, and the final decision is decided by majority voting. The confusion matrix is presented below.
>
> |                     | Morpheus pass | Morpheus discarded |
> | ------------------- | ------------- | ------------------ |
> | **Human pass**      | 42            | 3                  |
> | **Human discarded** | 17            | 9                  |
>
> We observe low false positives in the filtering module, meaning that valid videos are not being unnecessarily discarded. Instead, the remaining errors are actually predominantly false negatives (some invalid videos pass through), which implies that the filter is conservative: the reported discard rates are not inflated by pipeline fragility and may in fact underestimate the true rate of unusable generations. Consistently, **real-world videos yield essentially zero discard rate**, as expected for physically valid recordings under controlled conditions. Below are the the final metrics of our filtering pipeline.
>
> | Accuracy | Precision | Recall |
> | -------- | --------- | ------ |
> | 0.785    | 0.750     | 0.450  |
>
> **Q2: Failure mode analysis.** We agree that discard rates should be interpreted more carefully. Upon performing the visual inspection of the discarded samples, we found that the dominant failure mode is stillness. The other main categories are disappearance (object persistence failure) and duplication (visual generation failure where multiple copies of the object are spuriously created). By contrast, physical reasoning is evaluated only after this filtering stage. Since we additionally validate the trajectory pipeline and its robustness (see answer to Reviewer tzjf), these failure categories are not simply artifacts of tracking. We will clarify this breakdown more explicitly in the revision.
>
> **Q3: Real-world video results.** We apologize for the lack of clarity here. These results are reported explicitly in Figure 8 of Appendix A, where the per-experiment real-world results are shown. We will make this clearer in the main text.
>
> **Q4: Reporting individual metrics.** We agree that reporting only the averaged score in Figure 7 is incomplete. The per-experiment metric breakdown is provided in Appendix H, and we will make this reference explicit and improve the presentation in the revision.

---

> > ### Author Rebuttal · Reviewer_mnZD · 2026-04-04
> >
> > Thanks for the detailed rebuttal, my concerns are largely addressed. I will keep the original WA rating.

---

> > > ### Author Response · Authors · 2026-04-05
> > >
> > > Thank you for the thoughtful feedback and for engaging with our rebuttal — we appreciate your time and consideration.

---

### Official Review · Reviewer_tzjf · 2026-03-13

**Soundness:** 2
**Presentation:** 3
**Significance:** 4
**Originality:** 3
**Overall Recommendation:** 4
**Confidence:** 4

**Summary:**

This paper introduces Morpheus, a physics-informed evaluation framework for assessing whether video generative models (VGMs) respect Newtonian mechanics. The authors collect 130 real-world videos of controlled physical experiments (falling, projectile, pendulum, rolling, collision, spring, etc.) and propose two quantitative metrics. Results show that even the best models achieve combined scores well below real-world baselines, indicating that visual realism does not imply physical correctness.

**Compliance With Llm Reviewing Policy:**

Affirmed.

**Final Justification:**

The authors' rebuttal addressed my main concerns. I decide to raise my score.

**Key Questions For Authors:**

1. Can the existing paper paradigm be generalized to more general physics fields?

2. Can further analysis be provided on the impact of trajectory extraction error on the final score?

3. Regarding the choice of the PINN architecture, can additional ablation experiments be provided?

**If the above questions are properly answered, I am willing to increase my score.**

**Limitations:**

yes

**Strengths And Weaknesses:**

**Strengths:**
1. Evaluating the physical plausibility of video generation models is an urgent and crucial issue for the fields of video generation and embodied intelligence.

2. The proposed dynamics score and physical invariance score indices are based on known physical laws, rather than subjective judgments.

3. The analysis of failure modes further enhances the persuasiveness of the paper.

**Weakness:**

1. The research scope is limited to two-dimensional rigid-body Newtonian mechanics using a static camera. While acknowledging this, the authors do not discuss specific paths for extending to more general physics domains.

2. Relying on SAM2 for trajectory extraction may introduce additional illusions; the paper does not appear to systematically quantify the impact of external model errors on the final score.

3. The overall size of the dataset is relatively small, affecting the robustness of the evaluation.

---

> ### Author Rebuttal · Authors · 2026-03-31
>
> We thank the reviewer for the thoughtful feedback and for recognizing the importance of evaluating physical plausibility in video generation.
>
> **On generalization beyond the current setting.** We agree that the current benchmark is focused on controlled 2D rigid-body scenarios with a static camera. This restriction enables precise quantitative evaluation against known physical variables. We note that the framework is extendable beyond such setting. One concrete extension is an agentic data-construction pipeline that (i) curates candidate real-world videos, (ii) uses VLMs (e.g. Gemini 2.5 Pro) to generate scene descriptions/prompts, (iii) extracts object trajectories via our recognition and tracking pipeline, and (iv) retrieves and fits candidate governing equations from a physical-law library to instantiate new benchmark examples automatically. Such a pipeline would directly address the current manual components of our setup while scaling the benchmark to more diverse physical regimes. As one example, the same paradigm could be extended to hydrodynamics by collecting videos of objects moving in fluids, recovering their trajectories, and fitting candidate flow or reduced-order dynamical models to these observations. Related recent work on agentic scientific discovery [1] and visual equation discovery [2] provides evidence that multi-step automation of this kind is becoming increasingly realistic.
>
> **On trajectory extraction error.** We agree that quantifying the impact of the tracking pipeline on the final score is important. To address this, we are adding targeted robustness analyses that perturb the extracted trajectories and measure the resulting change in the physical score, as well as ablations of the trajectory-smoothing component. In particular, to evaluate the sensitivity to the common tracking errors like temporal gaps and mask boundary inconsistency, we perform two perturbations: (i) add zero mean Gaussian noise to the trajectory points (*Jitter*) and (ii) remove a portion of trajectory points (*Gaps*). In particular, we pick real-world and generated (Veo 3 and Kling) trajectories from the projectile experiment, and run scoring with different noise scales ($\sigma = 0.5, 1.0$) for Jitter perturbation, and randomly delete up to 20% of the points for Gaps. We find that across all settings, the total score deviates  $\leq 3$% in relative terms. Our smoothing block (see Appendix C) helps reduce the score deviation, e.g., 2.6% to 1.8% for real-world videos, Jitter perturbation with $\sigma=1.0$. For the Gaps perturbation, we note that the difference is even lower and does not exceed 1%. We attribute this overall robustness to the sliding window, smoothing, and PINN training, each of which helps handle noise and gaps.
>
> **PINN architecture ablations.** We also agree that the current pipeline has several modeling choices. We run the ablation on the number of hidden layers (1 to 4) and hidden dimension (8 to 128) for falling ball, pendulum, and projectile experiments.
>
> | Physics experiment | Paper score | Best score | Worst score |
> | ------------------ | ----------- | ---------- | ----------- |
> | Falling ball       | 0.975       | 0.99       | 0.98        |
> | Pendulum           | 0.953       | 0.95       | 0.16        |
> | Projectile         | 0.875       | 0.86       | 0.09        |
>
> We notice that for simple dynamics like falling ball, the scores stay nearly flat even for a very small architecture (one hidden layer with hidden dimension of 8). For more complex dynamics like pendulum and projectile, the deviation is greater, and the score grows with the number of parameters. However, we note that the score plateus early, and the difference between the best-performing PINN and the one we used in the paper is negligible, validating that our pick for architecture has sufficient capacity.
>
> [1]. Yamada, Yutaro, et al. "The ai scientist-v2: Workshop-level automated scientific discovery via agentic tree search." arXiv preprint arXiv:2504.08066 (2025).
>
> [2]. Feng, Tao, et al. "Physics-grounded motion forecasting via equation discovery for trajectory-guided image-to-video generation." arXiv preprint arXiv:2507.06830 (2025).

---

> > ### Author Rebuttal · Reviewer_tzjf · 2026-04-04
> >
> > Thank you for the detailed rebuttal. This answers every doubt I had. I will raise my score

---

> > > ### Author Response · Authors · 2026-04-04
> > >
> > > Thank you for the updated assessment and for the opportunity to strengthen the paper through your feedback.

---

### Official Review · Reviewer_FZdp · 2026-03-15

**Soundness:** 3
**Presentation:** 3
**Significance:** 3
**Originality:** 3
**Overall Recommendation:** 4
**Confidence:** 3

**Summary:**

This paper proposes Morpheus, a real-world physical consistency evaluation framework for video generation models, aiming to assess whether these models truly capture Newtonian mechanics rather than merely producing videos that look plausible. The authors construct a dataset of controlled real-world physical experiments, covering scenarios such as free fall, projectile motion, collisions, pendulum motion, and spring oscillation, and systematically evaluate a range of open-source and closed-source video generation models through trajectory extraction, dynamical equation fitting, and conservation-law stability analysis. The experimental results show that, although current models achieve strong visual quality, they still exhibit substantial deficiencies in adhering to physical laws. Overall, the key contribution of this paper lies in providing a more interpretable, quantitative, and realistic evaluation paradigm for assessing the physical understanding of video generation models.

**Compliance With Llm Reviewing Policy:**

Affirmed.

**Key Questions For Authors:**

See the cons part (1) & (4).

**Limitations:**

yes

**Strengths And Weaknesses:**

pros

(1) It explicitly evaluates generated videos along two dimensions: dynamical equation fitting and conservation-law stability. Moreover, these metrics are grounded in interpretable physical quantities such as energy, period, length, momentum, and gravitational acceleration.

(2) Instead of relying on simulator-generated data, the authors build a collection of real recorded physical experiments with fixed cameras and controlled initial conditions.

(3) The empirical findings are concrete and practically informative. For example, the authors show that keyframe interpolation outperforms single-frame extrapolation, that multi-frame inputs significantly improve physical plausibility, and that different models exhibit different failure modes, such as remaining static or causing objects to disappear or duplicate.

cons

(1) The paper claims that VLM scoring is insufficient to prove that a model understands physical laws. If the authors could supplement this with some real test examples, it would better support this claim.

(2) The current physical modeling is still relatively idealized. For example, the Physical Invariance score involves designs such as sliding windows, normalization, thresholds, and scaling, while the PINN part also requires manually specifying the ODE form and hyperparameters. Therefore, I think the paper could further clarify whether the conclusions remain stable when these specific implementation choices are changed.

(3) The method relies on 2D trajectory extraction and a series of engineering assumptions. All of the metrics in the paper are built on object segmentation, tracking, velocity/acceleration estimation, and 2D projection approximation.

(4) The authors do discuss existing baselines (in Table 2) and compare them with Morpheus at the level of related work discussion and benchmark attributes. They also provide conceptual and case-based explanations of the limitations of trajectory matching and VLM evaluation. However, they do not quantitatively compare the scores of these baseline evaluators with those of Morpheus on the same set of generated videos.

---

> ### Author Rebuttal · Authors · 2026-03-31
>
> We thank the reviewer for the thoughtful feedback and constructive suggestions.
>
> **Comparison with VLM-based and trajectory matching baselines.** We agree that our claim regarding the insufficiency of existing evaluators was not supported strongly enough in the current version. We added a quantitative comparison on our data with a current state-of-the-art VLM-based evaluator (VideoPHY2-AUTOEVAL [1]) and trajectory-matching metrics (PhysicsIQ [2]).
>
> For VideoPHY2, uses *Physical Commonsense* (PC) metric, a 1-5 scale rating of intuitive physical plausibility (1: Very Unlikely, 5: Very Likely), and *Physical Rules* (PR) metric, an indicator whether specific candidate rules (e.g. gravity, mass conservation, etc.) are followed (1), violated (0), or cannot be determined (2). Following VideoPHY2, we combine these metrics into *Joint Physical Consistency Score* (JPC), and indicator of whether the video is physically sound ($PC \ge 4$) and the specific physical rule is explicitly followed ($PR = 1$). Below we report the scores, as well as 95% confidence intervals.
>
> | Source         | Mean PC Score (1-5) | PR Adherence Rate (PR=1) | JPC Pass Rate        |
> | -------------- | ------------------- | ------------------------ | -------------------- |
> | **Real-World** | 3.71 [3.57, 3.85]   | 58.5% [49.8%, 67.2%]     | 23.6% [16.1%, 31.1%] |
> | **Generated**  | 3.53 [3.49, 3.56]   | 51.1% [48.5%, 53.8%]     | 13.9% [12.1%, 15.8%] |
>
> VideoPHY2-AUTOEVAL fails to assign near-ceiling scores to real ground-truth physics (e.g., PC $ = 3.71$ vs an expected $\approx$ 5.0) and only weakly separates real from generated videos, with no statistically significant difference in PR adherence and an extremely low JPC pass rate for real-world videos (**23.6%**). Failures are even more severe in specific cases, including misranking complex dynamics (double pendulum PC, 3.46 generated > 3.30 real-world), penalizing real sliding books/holonomic pendulums in PR while rewarding generated (82.8% generated vs. 40.0%), and near-total breakdown on common phenomena with 0% real-world JPC (projectiles, bouncing balls, sliding books, collisions, rolling cans). This suggests that while task-specific fine-tuning may improve performance, off-the-shelf VLM scoring is currently too brittle for precise physical evaluation.
>
> Regarding **trajectory matching metrics**, any kind of conditioning in VGMs leaves a number of physical parameters unobservable (e.g. mass distribution, friction coefficients, etc.). As a result, exact trajectory-matching metrics unfairly penalize valid but divergent trajectories. We calculated the Physics-IQ [2] metrics comparing two distinct, **real-world** recordings of the same physical phenomena, with similar, but slightly different initial conditions. The table reports the average metrics extracted from cross-comparing all real-world pairings per category.
>
> | Physical Experiment | Spatiotemporal IoU | Spatial IoU | Weighted Spatial IoU |
> | ------------------- | ------------------ | ----------- | -------------------- |
> | Bouncing Ball       | 0.100              | 0.274       | 0.169                |
> | Collision           | 0.021              | 0.184       | 0.112                |
> | Double Pendulum     | 0.087              | 0.561       | 0.408                |
> | Falling             | 0.164              | 0.396       | 0.342                |
> | Holonomic Pendulum  | 0.072              | 0.532       | 0.397                |
> | Projectile          | 0.010              | 0.040       | 0.034                |
> | Rolling             | 0.306              | 0.773       | 0.725                |
> | Sliding Book        | 0.189              | 0.240       | 0.202                |
> | Spring              | 0.189              | 0.298       | 0.233                |
>
> Interactive trajectories and scores per pair are visualized here: **[anonymous demo](https://anonymous.4open.science/w/dtoemq1iw0/)**
>
> Even perfect (real-world) physical events diverge due to minor unobservable differences. Trajectory matching evaluates a single possible pixel-aligned rollout against a single ground truth rather than adherence to physical laws, failing across multiple categories.
>
> **Robustness to implementation choices.** We agree that the current pipeline contains multiple components and relies on a relatively controlled setting. We added targeted ablations and perturbation-based analysis (see responses to Reviewers tzjf, mnZD, Xw1q); testing the score sensitivity to trajectory-processing errors, verifying key findings remain unchanged. We also outline extending our framework to broader domains for future work (see tzjf).
>
> [1]. Bansal, Hritik, et al. "Videophy-2: A challenging action-centric physical commonsense evaluation in video generation." arXiv preprint arXiv:2503.06800 (2025)
>
> [2]. Motamed, Saman, et al. "Do generative video models understand physical principles?." Proceedings of the IEEE/CVF Winter Conference on Applications of Computer Vision (2026)

---

### Decision · Program_Chairs · 2026-04-30

**Decision:**

Accept (regular)

**Comment:**

This paper introduces Morpheus, a physics-informed evaluation framework that assesses whether video generative models respect Newtonian mechanics, using real-world videos of controlled physical experiments and metrics based on conservation laws and governing ODEs.

Strengths:
- Addresses a timely and important question (physical plausibility of VGMs) with principled, interpretable metrics grounded in physical laws
- Uses real-world recorded experiments rather than synthetic simulations
- Comprehensive evaluation of 7 open/closed-source models under multiple conditioning strategies
- Strong rebuttal addressing all major concerns: added VLM baseline comparison (showing VLM scoring is insufficient), human correlation analysis (Spearman 0.708), robustness analysis under trajectory perturbation (<2.6% score deviation), PINN architecture ablation, and sliding window sensitivity analysis showing stable model rankings

Weaknesses (minor, acknowledged by authors):
- Limited to 2D rigid-body Newtonian mechanics with a static camera
- Modest dataset size (130 experiments)
- Finding that VGMs lack physics understanding is expected; the contribution is the evaluation methodology

Recommendation: Accept. The reviewers are in consensus, the rebuttal was thorough, and the framework fills a clear gap between visual quality benchmarks and physics-grounded evaluation.

Revision requests for camera-ready:
- Revise the title: "probing" has a specific technical meaning in ML (training classifiers on frozen representations to test encoded information). This paper evaluates generated outputs against physical laws, which is better described as "evaluating" or "benchmarking."
- Include the VLM comparison (VideoPHY2-AUTOEVAL) and trajectory matching analysis (Physics-IQ) from the rebuttal in the main paper or appendix
- Include the human correlation analysis and robustness perturbation results
- Correct the metric definition typo on Page 20 (noted by Reviewer Xw1q)
- Make real-world baseline scores (0.96-0.99 dynamical, ≥0.90 invariance) and per-experiment breakdowns more prominent in the main text
- Fix reference errors: (a) Cho et al., 2024 (Sora AGI survey) lists only 8 of 16 real authors with no "et al."; Puspitasari is the co-first author listed first on arXiv; (b) Kingma, 2013 is missing co-author Max Welling; (c) Liu et al. (Grounding DINO) and Xing et al. (DynamiCrafter) cited as 2025 but are ECCV 2024; (d) Wan et al. 2025a/b uses model/team name as first author ("Wan, T." is a BibTeX artifact of "Team Wan"), and is a duplicate entry; (e) Brooks 2024a/b duplicate; (f) Intelligence, P. (pi_0.5) and Team, G. R. (Gemini Robotics) are BibTeX artifacts of the organization names "Physical Intelligence" and "Gemini Robotics Team"; (g) Weng et al. (ART-V) was a CVPR 2024 Workshop paper, not the main conference; (h) Weissenborn et al. 2020 has Uszkoreit/Täckström in the wrong order